# CD36 inhibits β-catenin/c-myc-mediated glycolysis through ubiquitination of GPC4 to repress colorectal tumorigenesis

Yuan Fang [1,7], Zhi-Yong Shen[2,7], Yi-Zhi Zhan[3,4], Xiao-Chuang Feng[2], Ke-Li Chen[5], Yong-Sheng Li [2], Hai-Jun Deng[2], Su-Ming Pan[6], De-Hua Wu[1] & Yi Ding[1]

The diverse expression pattern of CD36 reflects its multiple cellular functions. However, the roles of CD36 in colorectal cancer (CRC) remain unknown. Here, we discover that CD36 expression is progressively decreased from adenomas to carcinomas. CD36 loss predicts poor survival of CRC patients. In CRC cells, CD36 acts as a tumor suppressor and inhibits aerobic glycolysis in vitro and in vivo. Mechanically, CD36-Glypcian 4 (GPC4) interaction could promote the proteasome-dependent ubiquitination of GPC4, followed by inhibition of β-catenin/c-myc signaling and suppression of downstream glycolytic target genes GLUT1, HK2, PKM2 and LDHA. Moreover, disruption of CD36 in inflammation-induced CRC model as well as $Apc^{Min/+}$ mice model significantly increased colorectal tumorigenesis. Our results reveal a CD36-GPC4-β-catenin-c-myc signaling axis that regulates glycolysis in CRC development and may provide an intervention strategy for CRC prevention.

[1] Department of Radiation Oncology, Nanfang Hospital, Southern Medical University, Guangzhou, Guangdong Province 510515, China. [2] Department of General Surgery, Nanfang Hospital, Southern Medical University, Guangzhou, Guangdong Province 510515, China. [3] Department of Pathology, Nanfang Hospital, Southern Medical University, Guangzhou, Guangdong Province 510515, China. [4] Department of Pathology, School of Basic Medical Sciences, Southern Medical University, Guangzhou, Guangdong Province 510515, China. [5] Huiqiao Medical Center, Nanfang Hospital, Southern Medical University, Guangzhou, Guangdong Province 510515, China. [6] Department of Radiation Oncology, Yue Bei People's Hospital, Shaoguan, Guangdong Province 512025, China. [7] These authors contributed equally: Yuan Fang, Zhi-Yong Shen. Correspondence and requests for materials should be addressed to D.-H.W. (email: 18602062748@163.com) or to Y.D. (email: dingyi197980@126.com)

CD36 is a membrane glycoprotein and expressed on platelets, macrophages, adipocytes, hepatocytes, myocytes, and some epithelia[1]. CD36 is involved in diverse cellular functions such as lipid metabolism, inflammatory response, clearance of apoptotic cells, and so on[2]. The fatty acid uptake function of CD36 was specifically correlated with its metastasis promoting role in a large number of human tumors. In oral squamous cell carcinoma, melanoma and luminal A breast cancer, CD36-positive proportions were highly predisposed to induce metastasis[3]. In ovarian cancer, cervical cancer, hepatocellular carcinoma, gastric cancer, elevated fatty acids absorption by CD36 could drive cancer progression and metastasis[4–7]. However, there seemed to exist more complex roles for CD36 regarding tumor development and progression. In glioblastoma, CD36 expression on cancer stem cells[8] or endothelial cells[9,10] could display totally different regulations in cancer progression. In breast cancer, the growth and aggressiveness-initiating[3,11] or -inhibiting[12–14] functions of CD36 have been solidly reported, and the paradoxical regulations were also seen in pancreatic adenocarcinoma[15,16]. In colorectal cancer, early evidence showed that decreased stromal expression of CD36 was positively correlated with vascularization as a receptor of thrombospondin 1[17], and a recent work has mentioned lncRNA TINCR could inhibit proliferation and promote apoptosis by activating CD36 in CRC cells[18], but the exact functional roles of CD36 in CRC development remain undefined.

Reprogramming of glucose metabolism is primarily glycolytic even in the presence of abundant oxygen in cancer cells. Aerobic glycolysis (the Warburg effect) has been shown to be a main consequence of oncogenic drivers[19]. Oncogenes like c-myc, mTOR and hypoxia-inducible factor 1α (HIF-1α) increase glycolytic activity to fulfill anabolic demands to sustain highly proliferating cancer cells[20–22]. Under normoxia, c-myc is a "master regulator" of glycolysis and tumor proliferation[23,24]. c-myc can transcriptionally upregulate genes encoding glucose transporter 1 (GLUT1)[25], lactate dehydrogenase A (LDHA)[26], hexokinase 2 (HK2)[27] and pyruvate kinase isoform 2 (PKM2)[28], thus promoting the activity of glycolysis with increased glucose uptake and fast conversion of glucose to lactate.

Colorectal cancer (CRC) ranks the third most common cancer and accounts for the second leading cause of cancer mortality worldwide[29,30]. Canonical Wnt signaling plays crucial roles in regulation of both normal intestinal maturation and colorectal tumorigenesis[31]. Activation of β-catenin is the key nuclear effector of Wnt signaling. Increased cytoplasmic and nuclear translocation of β-catenin, followed by interacting with members of the T-cell factor (TCF) family, leading to increased cell proliferation and growth[32]. c-myc is a well-established target gene of β-catenin/TCF transcription factor complex, aberrantly nuclear accumulation of β-catenin and constitutive upregulation of c-myc are believed to be the basis of colorectal tumorigenesis[33]. Although Wnt signaling has been reported to regulate glycolysis in colon cancer[34], the mechanism involves CD36-mediated β-catenin/c-myc signaling in controlling glycolysis in CRC has never been discussed to our knowledge.

Here, we report a progressively decreased expression and a tumor-suppressive role of CD36 in CRC development. Moreover, our results provide new mechanistic insights into the crucial roles of CD36 in suppressing β-catenin/c-myc signaling via promoting the proteasome-dependent ubiquitination of GPC4, which result in subsequent inhibition of downstream aerobic glycolysis and colorectal tumorigenesis.

## Results

**Expression and prognosis of CD36 in CRC development**. To explore the expression of CD36 in colorectal tumorigenesis, we firstly analyzed CD36 messenger RNA (mRNA) expression with the public GEO (Gene Expression Omnibus) databases (GDS2947, GSE10950, GSE44076, GSE74602, GDS3756, GSE11223) and TCGA (The Cancer Genome Atlas) database, results revealed that CD36 mRNA level was downregulated independently in inflamed location of ulcerative colitis, adenomas and tumors compared with their normal counterparts (Fig. 1a and Supplementary Fig. 1a). Consistent with public data, mRNA and protein levels of CD36 by quantitative real-time polymerase chain reaction (qRT-PCR) and western blot analysis were similarly decreased in colorectal tumors as compared with their paired normal tissues (Fig. 1b, c). Additionally, there was a progressive loss in CD36 expression from adenomas to carcinomas (Fig. 1d and Supplementary Fig. 1c), and we also found an intensive loss of CD36 in adenomas with malignant transformation (Supplementary Fig. 1b), so we speculated that CD36 loss might be an early event in CRC development and associated with the malignant transformation of adenomas.

Next, we investigated the clinical relevance of CD36 expression in a tissue microarray (TMA) consisting of 90 pairs of human colon cancer samples by immunohistochemistry (IHC), and finally 81 samples with visible tumor epithelium were included for following analysis. Weak or absent CD36 expression was seen in almost two-thirds tumor samples examined, whereas almost all normal specimens showed strong CD36 signal (Fig. 1e). Kaplan–Meier curves showed negative CD36 expression predicted poorer overall survival than those with positive CD36 staining (Fig. 1f). Although there was no significant correlation between CD36 and clinicopathologic characters such as age, gender, tumor size, differentiation or pathological stage (Supplementary Table 1), univariate and multivariate Cox regression analysis showed negative CD36 expression was associated with increased risk of death and was an independent risk factor for poorer overall survival after adjustment for risk factors including age and AJCC stage in patients with colon cancer, respectively (Supplementary Table 2). Additionally, relatively low CD36 mRNA expression also predicted worse disease-free survival in GSE24551 dataset with stage II and stage III CRC patients involved (Fig. 1f). Collectively, these results suggested that CD36 is progressively downregulated in CRC development and loss of CD36 correlates with an unfavorable prognosis of CRC patients.

**CD36 plays anti-carcinogenic roles in CRC**. Given previous results, we further evaluated the functional role of CD36 in CRC cells. As compared with normal colonic mucosa cell line FHC, CD36 expression was decreased in almost all CRC cell lines as determined by qRT-PCR and western blot analysis (Supplementary Fig. 2a). RFP-tagged CD36-expressing recombinant lentivirus (LV-CD36) or control vectors (LV-RFP) were used to establish CD36-overexpressed SW480 and LoVo cells, while CD36-targeting shRNA (shCD36) or corresponding controls (shNC) were introduced into RKO and CACO2 cells with relatively high endogenous CD36 expression. Ectopic CD36 expression in SW480 and LoVo cells significantly inhibited cell growth and colony formation, while CD36 knockdown promoted both of them (Fig. 2a and Supplementary Fig. 2b). We next evaluated the effect of CD36 on apoptosis. Results showed LoVo cells displayed a higher apoptosis rate after CD36 overexpression under normal conditions, while both SW480 and LoVo cells with CD36 overexpression showed a significant increase of apoptosis as compared with their negative controls when under 5-fluorouracil (5-Fu) treatment. Conversely, cells with shCD36 showed sharply less apoptosis than their negative controls when treating with 5-Fu. Consistently, protein expression of apoptosis markers, like cleaved-caspase-3, cleaved-caspase-9 were upregulated and Bcl-

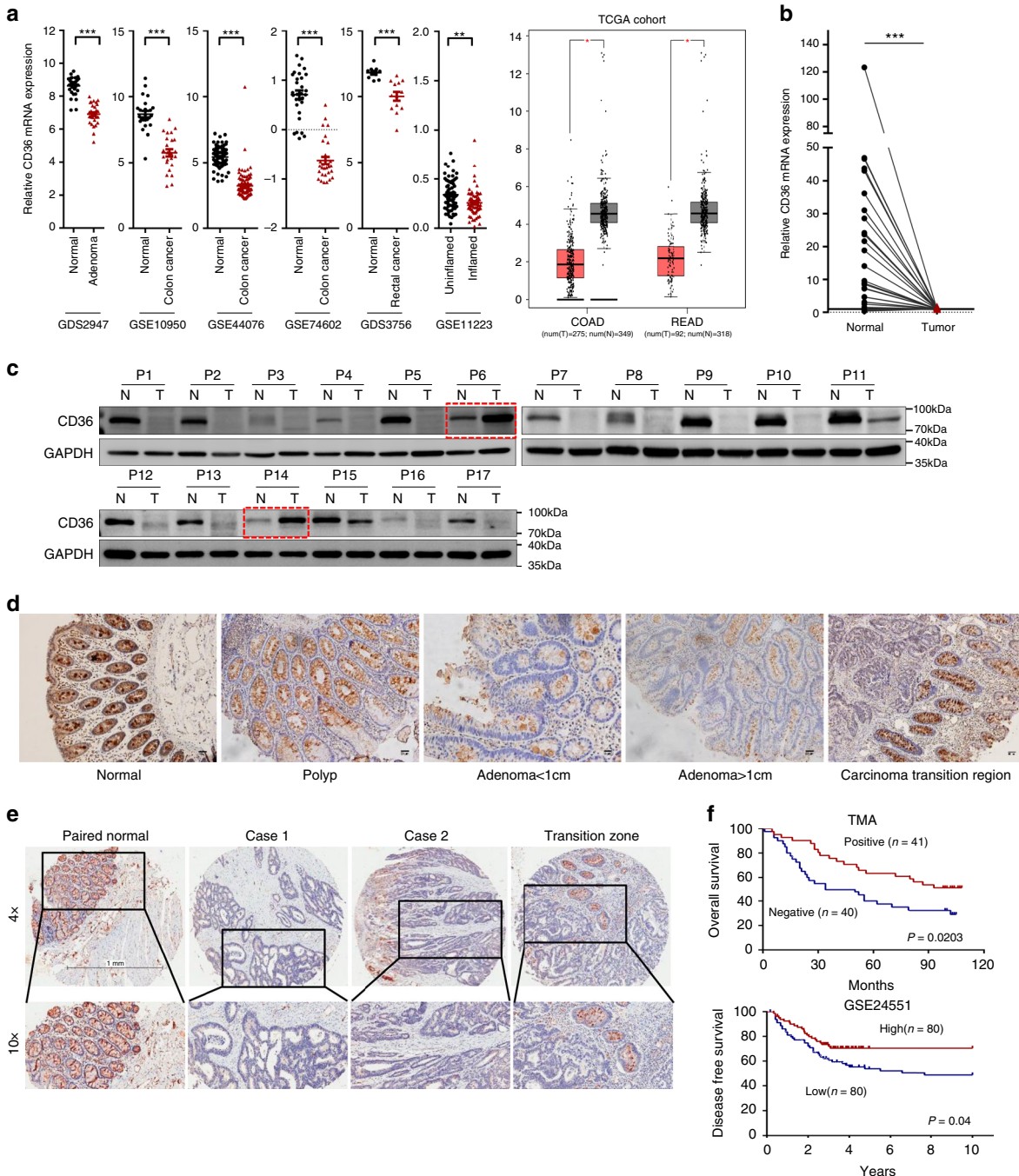

**Fig. 1** CD36 in the development of CRC. **a** Gene chip data were obtained and compared from 6 GEO datasets and TCGA cohorts. COAD colon adenocarcinoma, READ rectal adenocarcinoma. Results are shown as mean ± SEM, **P < .01, ***P < .001, based on Student's t-test. **b** qRT-PCR analysis of mRNA expression in paired CRC tissues. Results are shown as mean ± SEM (n = 35), ***P < .001, based on paired Student's t-test. **c** Western blots of CD36 protein in 17 pairs of CRC tissues, GAPDH was loaded as a control. **d** Immunohistochemistry (IHC) of CD36 in colon tissues with different stages of lesions, Scale bar, 50 μm (20×). **e** Representative images of CD36 staining on tissue microarray (inserts show ×2.5 magnification), Scale bar, 1 mm. **f** Kaplan–Meier survival curves of TMA (n = 81) and GSE24551 (n = 160) analysis. Source data are provided as a Source Data file

XL was downregulated in cells with LV-CD36, and opposite changes on these key apoptosis markers were observed in cells with shCD36 (Fig. 2b and Supplementary Fig. 2c). On cell cycle regulation, CD36 overexpression resulted in cell cycle arrest while knockdown of CD36 promoted cell cycle progression in CRC cells, and western blot analysis showed different CD36 status had opposite regulations on cell-cycle governors, like Cyclin D1, CDK4, CDK6, p21 and p18 (Fig. 2c and Supplementary Fig. 2d). Moreover, transwell assays revealed CD36 could significantly

decrease the migratory and invasive ability of SW480 cells, while CD36 knockdown promoted both of them in RKO cells (Supplementary Fig. 2e). Taken together, these data supported the tumor-suppressive roles of CD36 in CRC cells.

**CD36 represses aerobic glycolysis in CRC.** To explore CD36-mediated molecular events, we performed an iTRAQ coupled with LC-MS/MS to compare the proteome profiles of SW480

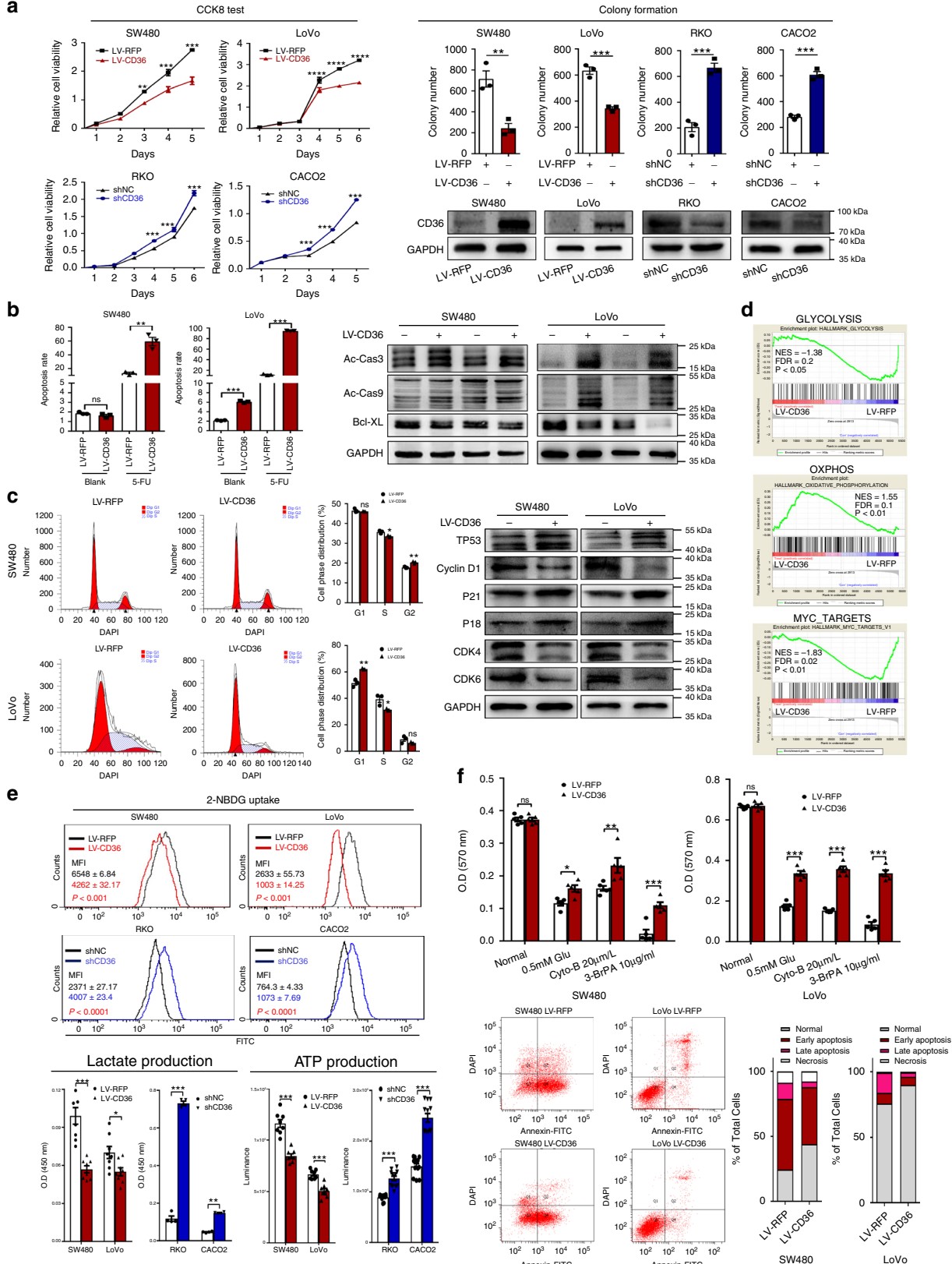

**Fig. 2** CD36 plays anti-carcinogenic roles via repressing glycolysis. **a** CCK8 assays and colony formation assays, transfection efficiency was determined by western blot. **b** Cell apoptosis under normal conditions and 5-Fu treatment and western blots of apoptosis markers. **c** Cell cycle analysis and western blots of cell cycle markers. **d** Gene-set enrichment analysis (GSEA) of the protein profiles between SW480 LV-CD36 and SW480 LV-RFP cell lines. **e** 2-NBDG uptake, lactate production and ATP production. **f** Cell viability was measured by MTT assays (up) under different culture conditions. Cell apoptosis of SW480 and LoVo cells (LV-RFP vs. LV-CD36) with low glucose conditions (down). Each experiment was performed at least triplicate and results are presented as mean ± SEM. Student's *t*-test, one-way ANOVA or two-way ANOVA was used to analyze the data (*$P$ < .05, **$P$ < .01, ***$P$ < .001, ****$P$ < .0001). Source data are provided as a Source Data file

LV-CD36 and LV-RFP cells by gene-set enrichment (GSEA) analysis. Among the hallmark signature gene sets, control group was found to strongly associate with MYC targets and glycolysis, while LV-CD36 group positively correlated with oxidative phosphorylation (OXPHOS) (Fig. 2d), suggesting a metabolic shift from aerobic glycolysis to OXPHOS due to CD36 overexpression in CRC cells. Moreover, GSEA analysis of GEO RNA sequencing results of CRC patients (GSE74602, GDS2947, GDS3756) also indicated that relatively low expressions of CD36 were significantly associated with MYC targets and glycolysis (Supplementary Fig. 3a). As MYC is a key regulator of glycolysis in tumors, we hypothesized that CD36 might play an important role in regulating MYC-mediated glucose metabolism in CRC.

We firstly tested whether CRC cells with different CD36 expression would exhibit altered glycolytic activities. Our results showed that CD36 overexpression in SW480 and LoVo cells displayed significantly decreased 2-NBDG uptake, lactate release, and ATP production in comparison with their negative controls, while the glycolytic repression was significantly reversed in RKO and CACO2 cells with CD36 knockdown (Fig. 2e). We then cultured cells under low glucose conditions (0.5 mM glucose), with cytochalasin B (Cyto-B, a glucose transport inhibitor, 20 μM)[35], or with 3-bromopyruvic acid (3-BrPA, a hexokinase-2 inhibitor, 10 μg/ml)[36] for 48 h, respectively. Results showed that CD36-overexpressed cells were relatively resistant to the deprivation of glucose or inhibition of glycolysis, as manifested by less cell death and apoptosis in CD36-overexpressed cells than were in control cells (Fig. 2f). Using colony formation assays, we further demonstrated that cells with CD36 knockdown were highly addicted to glucose, and gradient deprivation of glucose significantly inhibited their viability, while CD36-overexpressed cells showed significantly less glucose addiction than their negative controls (Fig. 3a and Supplementary Fig. 3b). Based on our data, we hypothesized that loss of CD36 might turn on aerobic glycolysis in CRC cells.

**CD36 inhibits β-catenin/c-myc-mediated glycolysis**. As GSEA analysis showed loss of CD36 was positively correlated with MYC targets, we then sought to examine the expression of MYC and its downstream glycolytic targets in CRC cells. Firstly, we found the mRNA levels of CD36 were negatively correlated with MYC expression in GEO and TCGA datasets with gene chip data from CRC patients (Supplementary Fig. 3c). We then revealed that both mRNA and protein level of c-myc and its downstream glycolytic targets including GLUT1, HK2, LDHA and PKM2 were downregulated in SW480 and LoVo with LV-CD36 compared to LV-RFP cell lines. Conversely, knockdown of CD36 resulted in marked increase in their expressions (Fig. 3b). As activation of Wnt signaling due to APC mutation is often the first detectable molecular event in early colorectal adenomas[31], and c-myc is a major transcriptional target of β-catenin and is often strongly overexpressed in CRC[37], so we wondered if CD36 could regulate the expression and/or location of β-catenin. In keeping with our hypothesis, the protein level of β-catenin was downregulated after CD36 overexpression but was upregulated in CD36-deficiency cells by western blot analysis (Fig. 3b). Moreover, immunofluorescence (IF) analysis showed that in control SW480 and LoVo cells, β-catenin was mainly located in the nucleus and cytoplasm and had a moderate co-localization with nuclear DAPI, whereas ectopic CD36 expression resulted in increased cytoplasmic membrane translocation of β-catenin and significantly weakened the nuclear co-localization between β-catenin and DAPI (Fig. 3c and Supplementary Fig. 3d).

To investigate whether β-catenin and c-myc activation are responsible for CD36-deficiency-regulated glycolysis, we used

pharmacologic approaches to inhibit their expression, respectively. As expected, knockdown of CD36 rendered CRC cells much more sensitive to both β-catenin inhibitor XAV-939[38] and c-myc inhibitor 10058-F4[39], as indicated by more severe repression of the protein expression of downstream glycolytic genes and more cell death in RKO and CACO2 cells with shCD36 than were in control cells (Fig. 3d, e). As MYC and Cyclin D1 are two well-established transcriptional targets of β-catenin, we further conducted chromatin immunoprecipitation (ChIP) assays to examine the in vivo association of nuclear β-catenin with TCF-binding elements (TBEs) located within the promoters of MYC and Cyclin D1. Results showed CD36 overexpression significantly decreased the β-catenin to the promoters of MYC and Cyclin D1 (Fig. 3f). Collectively, these results implied that CD36 contributed to the transcriptional inhibition of β-catenin/c-myc signaling-mediated glycolysis in CRC cells.

**CD36 induces proteasome-dependent ubiquitination of GPC4**. To identify the molecular mechanistic basis by which CD36 inhibits β-catenin/c-myc signaling, we analyzed our iTRAQ data and found Glypican 4 (GPC4) was the most significantly down-regulated protein after CD36 overexpression (Supplementary Fig. 4a). In light of these findings, we then evaluated the protein levels of GPC4 and found that GPC4 was significantly suppressed in CD36-overexpressed cells, whereas increased GPC4 levels were seen in cells with CD36 knockdown (Fig. 4a). Interestingly, the mRNA did not show consistent changes (Supplementary Fig. 4b). Analysis about TCGA database revealed no significant difference of GPC4 mRNA expression between tumors and their normal counterparts (Supplementary Fig. 4c), but our IHC results of 10 randomly selected transition zones from CRC patients revealed that the protein level of GPC4 was significantly upregulated in tumor locations as compared with their adjacent normal mucosa (Supplementary Fig. 4d). Hence, we wondered whether CD36-mediated GPC4 downregulation involved post-transcriptional modification.

We first analyzed the subcellular location of CD36 and GPC4 by IF and found CD36 signal significantly overlapped with GPC4 on the cellular membrane and cytoplasm in different CRC cells (Fig. 4b and Supplementary Fig. 4e). To gain insight into the potential interaction between CD36 and GPC4, Co-immunoprecipitation (Co-IP) and western blot were performed by comparing the anti-Flag IP product of SW480 and LoVo LV-CD36 cell lysates with anti-IgG IP product, and GPC4 was found to interact with CD36. To further validate the endogenous interaction of them, Co-IP using anti-CD36 antibody was incubated with RKO and CACO2 cell lysates, and GPC4 could also be co-precipitated by CD36 in both cell lines. Reciprocal Co-IP further confirmed the interaction between them using GPC4 antibody to co-precipitate CD36 in these cells (Fig. 4c and Supplementary Fig. 4f). As we know, ubiquitination is instrumental in the regulation of protein expression. In light of the reverse regulation of GPC4 by CD36, we firstly pretreated cells with cycloheximide (CHX) to determine the stability of GPC4. Results showed that the half-life periods of GPC4 were much longer in CRC cells with CD36 knockdown than were in control cells, whereas GPC4 protein degraded faster after CD36 over-expression (Fig. 4d and Supplementary Fig. 5a). Next, we treated cells with proteasome inhibitor MG132 or lysosome inhibitor 3-Methyladenine (3-MA), results revealed that MG132 treatment could significantly increase the GPC4 protein expression in control cells, while only a slight upregulation of GPC4 was observed in cells with CD36 knockdown (Fig. 4e), and 3-MA treatment did not affect the GPC4 protein level (Supplementary Fig. 5b), suggesting that CD36 might modulate GPC4 stability in

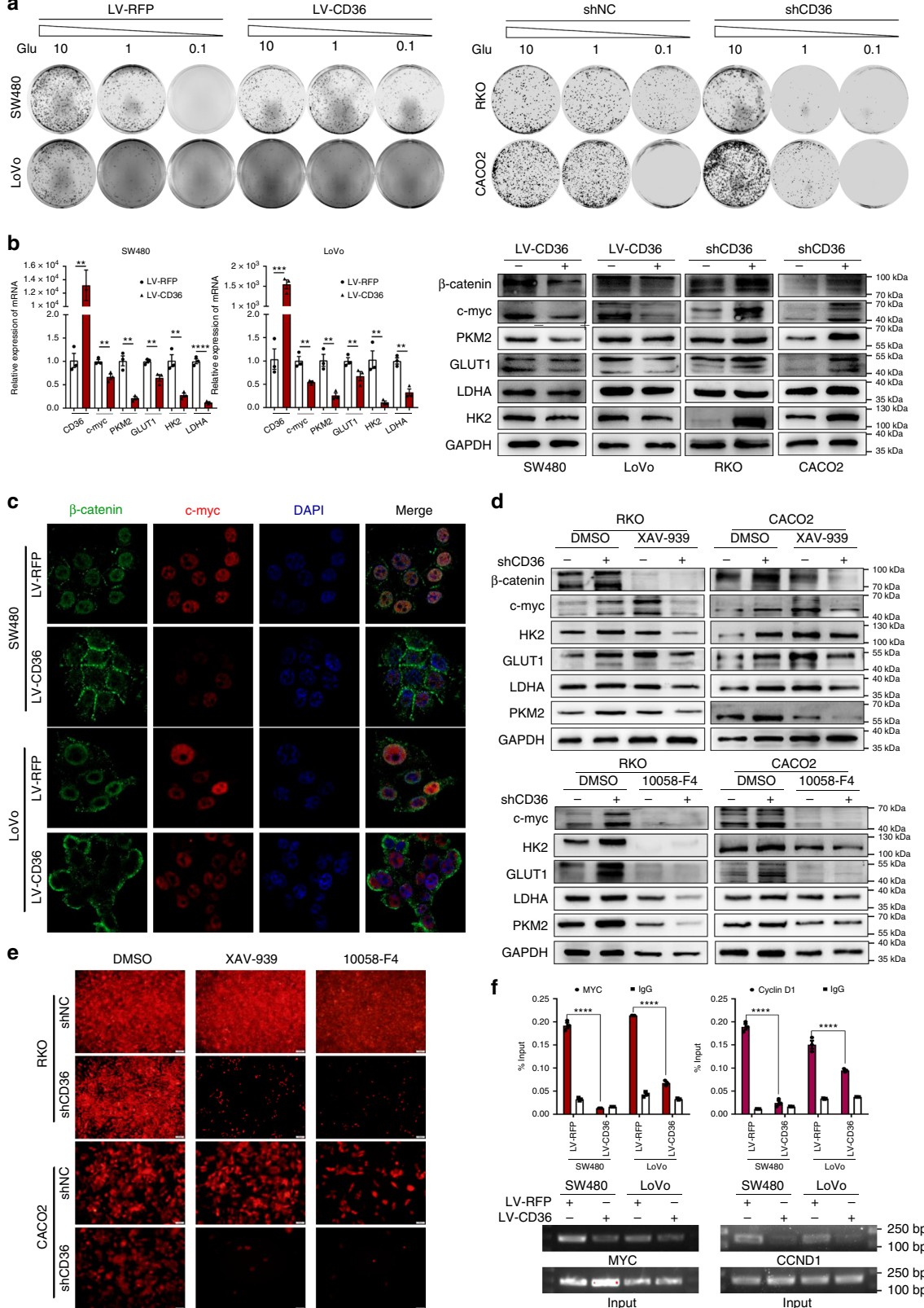

**Fig. 3** CD36 inhibits glycolysis via suppressing β-catenin/c-myc. **a** Colony formation assays to test cell viability under different glucose deprivation. **b** qRT-PCR and western blots of target genes in indicated cell lines. Results are shown as mean ± SEM ($n = 3$), **$P < .01$, ***$P < .001$, ****$P < .0001$, based on Student's $t$-test. **c** IF staining of β-catenin (green) and c-myc (red). Merged images represent overlays of β-catenin, c-myc and nuclear staining by DAPI (blue). **d** Western blots of indicated proteins in cells treated with β-catenin inhibitor XAV-939 (10 μM, 72 h) and c-myc inhibitor 10058-F4 (25 μM, 96 h), respectively. **e** Images of cells with shCD36 or shNC under treatment of XAV-939 (100 μM, 36 h) or 10058-F4 (200 μM, 36 h). Scale bar, 100 μm (10 ×). **f** ChIP-qPCR and PCR analysis of β-catenin with MYC and Cyclin D1 promoter regions. Data are shown as fold enrichment relative to input, ****$P < .0001$, mean ± SEM ($n = 3$), based on Student's $t$-test. Source data are provided as a Source Data file

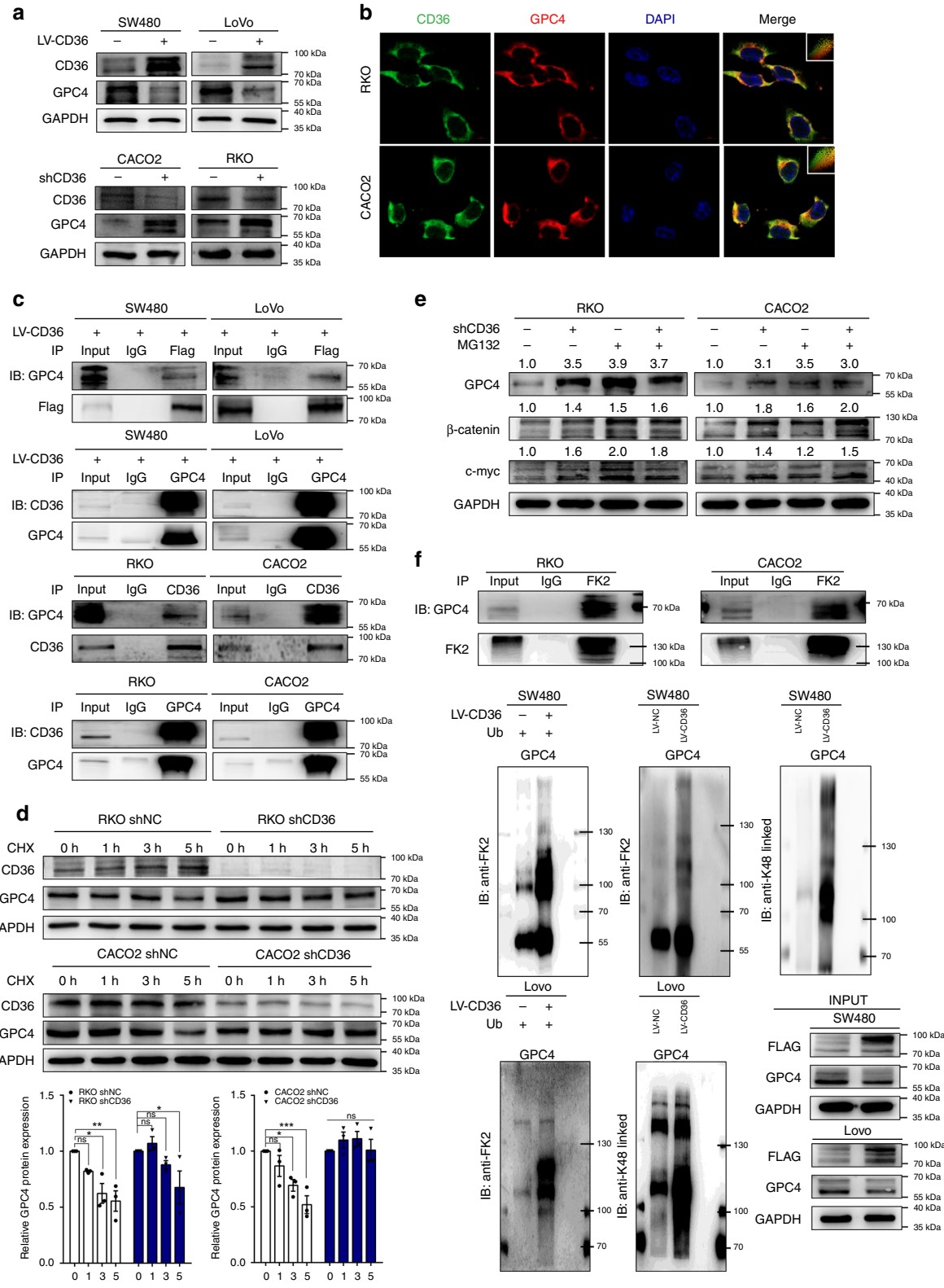

**Fig. 4** CD36 mediates proteasome-dependent ubiquitination of GPC4. **a** Protein levels of GPC4 in CRC cell lines with different CD36 expressions. **b** IF analysis of the co-localization of CD36 (green) and GPC4 (red) in RKO and CACO2 cell lines. **c** Co-immunoprecipitation (Co-IP) identified the interaction between CD36 and GPC4. **d** Time indicated cycloheximide (CHX, 20 μg/ml) treatment to compare the stability of GPC4 in indicated cell lines. Results are shown as mean ± SEM ($n = 3$), *$P < .05$, **$P < .01$, ***$P < .001$, based on two-way ANOVA. **e** MG132 (20 μM, 24 h) treatment on RKO and CACO2 cells with/without CD36 knockdown. **f** Co-IP determined endogenous ubiquitination and proteasome-dependent ubiquitination of GPC4. Source data are provided as a Source Data file

a proteasome-dependent manner. In support of the endogenous ubiquitination of GPC4, Co-IP using total ubiquitin FK2 antibody was then incubated with RKO and CACO2 cell lysates, results showed that GPC4 could be co-precipitated by FK2 in both cell lines. As K48-linked poly-ubiquitination generally targets proteins for proteasomal degradation[40], we further identified that overexpression of CD36 could increase K48-linked poly-ubiquitination of GPC4 and the interaction between GPC4 and FK2 in both SW480 and LoVo cells (Fig. 4f). Taken together, these results indicated that CD36-GPC4 interaction could induce GPC4 proteasome-dependent ubiquitination and degradation in CRC cells.

**The tumor-suppressive function of CD36 is dependent on GPC4.** It has been reported that GPC4 could activate Wnt3a–β-catenin pathway via promoting phosphorylation of LRP6 at Ser1490 [41], so we further tried to determine the role of GPC4 on β-catenin/c-myc signaling. Firstly, we found MG132 treatment could sharply increase the expression of β-catenin and c-myc in control cells, while only a slight upregulation of them were observed in cells with CD36 knockdown (Fig. 4e). In addition, MG132 treatment could upregulate their expressions either in cytoplasm or in nucleus, and the upregulations were more obvious in CD36-overexpressed cells than were in control cells (Fig. 5a and Supplementary Fig. 5c). Confocal microscopy images further confirmed the nuclear accumulation of β-catenin in CD36-overexpressed cells treated with MG132 (Supplementary Fig. 5d). These results strongly suggested regulation of ubiquitination played pivotal roles in CD36-mediated inhibition of β-catenin/c-myc signaling.

To test our idea, we generated GPC4-overexpressed plasmids to examine the importance of GPC4 in CD36-mediated tumor-suppressive function. Consistent with previous work, we firstly found GPC4 overexpression could lead to a higher upregulation of β-catenin after Wnt3a treatment than negative controls. Western blot analysis also identified a higher phosphorylation of LRP6 at Ser1490 and β-catenin expression in Wnt3a-treated GPC4-overexpressed 293T cells (Supplementary Fig. 5e). Then, our results showed ectopic expression of GPC4 significantly increased the protein levels of β-catenin, c-myc and downstream glycolytic targets in cells with/without CD36 overexpression (Fig. 5b). Immunofluorescence analysis further confirmed the nuclear accumulation of β-catenin after ectopic expression of GPC4 (Fig. 5c and Supplementary Fig. 5f). In addition, overexpression of GPC4 significantly increased the proliferation and colony formation ability of cells with different CD36 status (Fig. 5d, e). ATP production, glucose consumption and lactate release were also significantly increased by forced GPC4 expression (Fig. 5f). Hence, our results suggested GPC4 was indispensable for CD36-mediated tumor-suppressive functions in CRC cells.

**CD36 inhibits growth and metastasis of CRC cells in vivo.** In light of our in vitro findings, we tested the functions of CD36 in proliferation in vivo with subcutaneous xenograft models at first. Results showed SW480 LV-CD36 cells suppressed tumor growth by 63%, while knockdown of CD36 in RKO cells increased tumor growth by 76%, when compared to their negative controls, respectively. Consistent with in vitro findings, CD36 overexpression significantly inhibited cell proliferation in tumors as determined by Ki-67 staining, while CD36 depletion markedly enhanced proliferation (Fig. 6a and Supplementary Fig. 6a, b). Furthermore, CD36 overexpression led to significant downregulation of GPC4, β-catenin, c-myc and downstream glycolytic genes, including GLUT1 and LDHA (Fig. 6b). Then, we chose to perform intrasplenic

inoculation model to detect the role of CD36 in metastasis. Surprisingly, results revealed that no liver metastasis was found in all of the seven mice injected with CD36-overexpressed SW480 cells, while in the control group, metastatic lesions could be observed in the livers of all mice involved (Fig. 6c, d). Collectively, these results supported the view that CD36 played anti-carcinogenic roles in CRC cells in vivo.

**CD36 knockdown sensitizes CRC cells to glycolytic inhibition.** Since in vitro silencing of CD36 significantly upregulated glycolysis, we therefore wanted to investigate whether inhibition of glycolytic targets could effectively inhibit tumor growth with CD36 deficiency in vivo. Stable RKO cells transfected with either shNC or shCD36 were subcutaneously inoculated in nude mice, when palpable tumors were present in all animals, then mice were randomly divided into three groups and were treated with intraperitoneal injection of PBS or 3-BrPA (a HK2 inhibitor) or WZB117 (a GLUT1 inhibitor)[42] every 2 days, respectively. Results revealed that RKO cells with CD36 knockdown treated with 3-BrPA or WZB117 grew at a significantly slower rate than those in the PBS control group. A significant decrease in tumor volume was observed in shCD36 + 3-BrPA (76.7% reduction) and shCD36 + WZB117 (69.8% reduction) group as compared to the shCD36 + PBS group. However, drug delivery seemed to induce a growth promotion of the RKO shNC tumors, we thought it might due to a more supportive environment as drugs efficiently suppressed the tumor growth in the opposite side of the same mice, or maybe there involved other complicated mechanisms we were not able to figure them out yet (Fig. 6e and Supplementary Fig. 6c). Growth regression was further confirmed by quantifying Ki-67 staining, and protein expression of HK2 and GLUT1 were also analyzed to identify the drug effectiveness (Fig. 6f). These results suggested that blocking glycolysis might be a sensitive method in treating CRC with CD36 loss.

**AAV-mediated CD36 knockdown promotes AOM/DSS-induced CRC.** Recombinant adeno-associated virus (AAV) vector-mediated gene delivery to intestinal epithelial cells provides a new approach of gut transduction and allows the study of intestinal diseases[43]. According to paper reported, we selected AAV serotypes 9, which provides relatively high efficiency in gut transduction[44], and transfected 4-week-old male BALB/c mice with AAV-CD36 knockdown (AAV-CD36-KD) or control vectors (AAV-NC) through tail vein injection. To determine intestinal efficiency in vivo, western blots and IHC assays were used to detect GFP expression in intestines and colons (Supplementary Fig. 7).

Firstly, we used the AAV-mediated genetic approaches combined with AOM/DSS model to determine if CD36 loss contributed to inflammatory tumorigenesis (Supplementary Fig. 6e). Results showed although colonic tumors could be detected in almost whole colons (mainly in the distal colons) and rectums in both AAV-CD36-KD (100%) and AAV-NC mice (75%). Mice in AAV-CD36-KD group showed more tumors and larger tumor volumes compared to their control mice (Fig. 7a). Consistently, tumors formed in AAV-CD36-KD group showed significantly more proliferating cells as evidenced by higher proportions of Ki-67 and PCNA staining in both normal mucosa and tumor sections. In addition, IHC analysis showed CD36 depletion led to significant upregulation of GPC4, β-catenin and c-myc in tumors (Fig. 7b and Supplementary Fig. 6e). Collectively, these findings provided genetic evidence supporting our hypothesis that CD36 regulates GPC4-mediated Wnt activity in AOM/DSS-induced mice model.

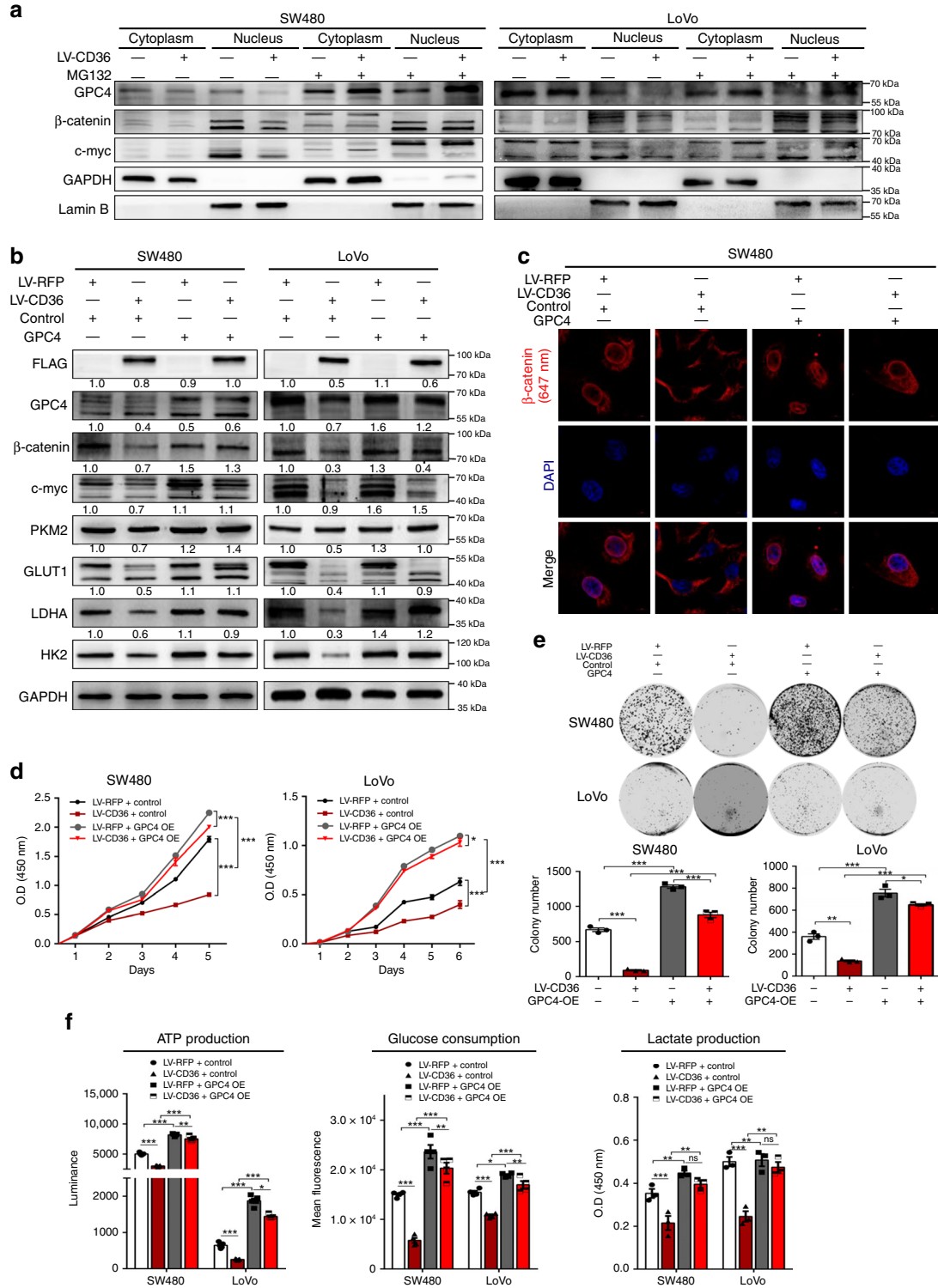

**Fig. 5** The functional role of CD36 is dependent on GPC4. **a** Western blot of indicated proteins with/without MG132 treatment. Lamin B is a nuclear marker, GAPDH was loaded as a cytoplasmic marker. **b** Western blots of indicated proteins in SW480 and LoVo cells (LV-RFP vs. LV-CD36) with/without ectopic expression of GPC4. **c** IF analysis of β-catenin (red) location after forced expression of GPC4. **d** CCK8 tests. **e** Colony formation assays. **f** ATP production (left), glucose consumption (middle) and lactate production (right). Each experiment was performed in at least triplicate and results are presented as mean ± SEM. Student's *t*-test or two-way ANOVA was used to analyze the data (*P < .05, **P < .01, ***P < .001, ****P < .0001). Source data are provided as a Source Data file

**CD36 loss promotes tumorigenesis in *Apc*^Min/+^ mice.** To further verify the regulatory features of CD36 in vivo, we introduced AAVs into the *Apc*^Min/+^ mice with vein injection and examined the tumor growth. Results showed inactivation of CD36 caused a

significant increase of tumor numbers in the large intestines, most tumors formed in AAV-CD36-KD *Apc*^Min/+^ mice showed much higher dysplasia with cauliflower-like uplift, while the mean diameter of tumors were not significantly different, perhaps due

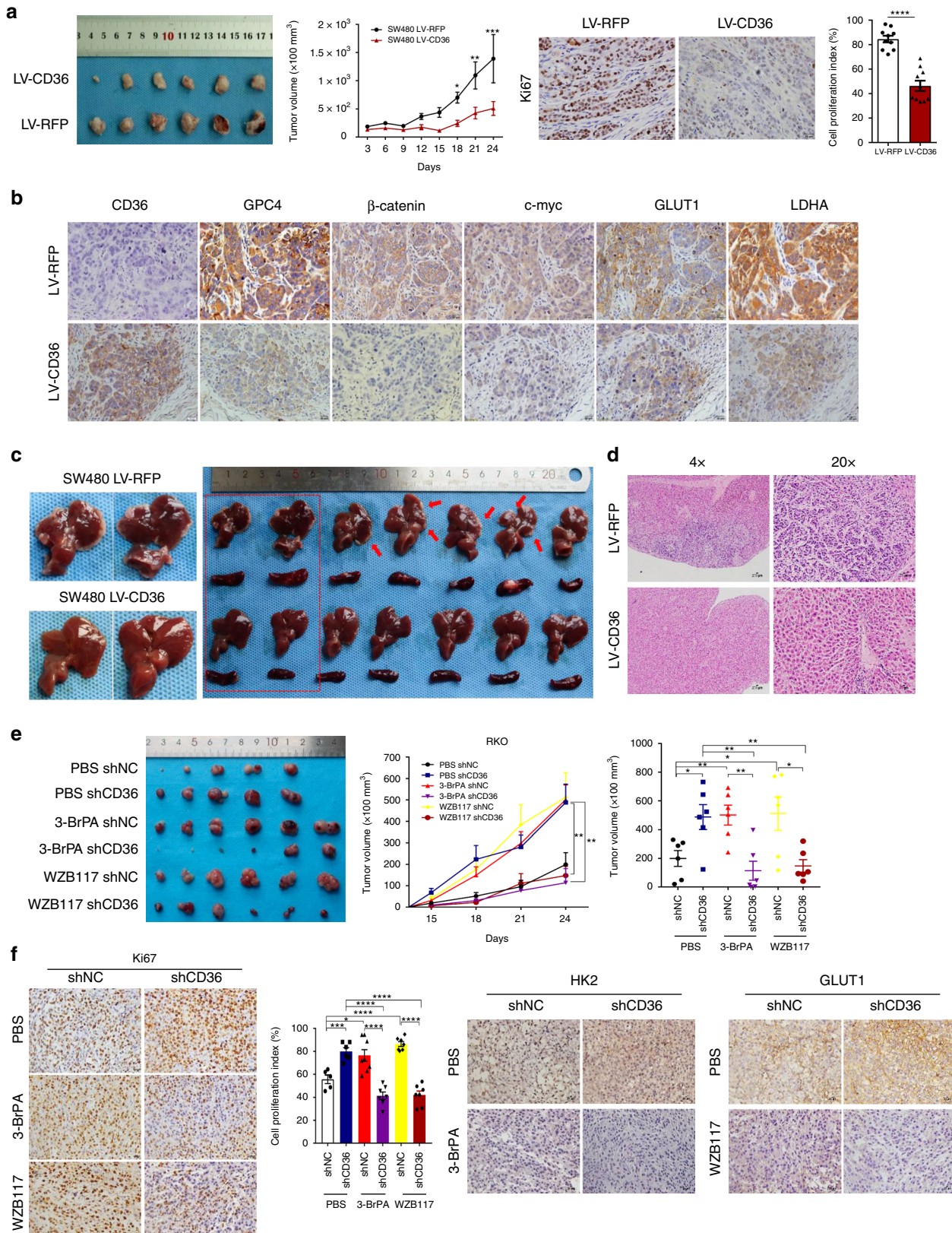

to our small sample size (Fig. 7c). The intensity of PCNA staining was significantly increased in both the normal and tumor sections of AAV-CD36-KD group than were in the control group (Fig. 7d). IHC results also showed a remarkably increased expression of GPC4, β-catenin, c-myc and downstream glycolytic genes in the tumors of AAV-CD36-KD mice (Fig. 7e). Taken together, these results further suggested the molecular mechanism by which CD36 controls tumor proliferation and glycolysis via inhibiting GPC4-mediated β-catenin/c-myc signaling in colorectal tumorigenesis (Fig. 7f).

**Fig. 6** CD36 plays tumor-suppressive roles in vivo. **a** Subcutaneous xenograft tumor growth in nude mice (6 per group) were measured and compared (left) in SW480 (LV-RFP vs. LV-CD36) cell lines. Cell proliferation index was determined by the proportion of nuclear Ki-67–positive cells (right). **b** Representative images of IHC staining of CD36, GPC4, β-catenin, c-myc, GLUT1 and LDHA on tumor sections. Scale bar, 20 μm (40×). **c** Macroscopic appearance of livers and spleens of mice with intrasplenic inoculation (7 per group). **d** Representative images of H&E staining of liver sections in CD36-overexpressed and control group, Scale bar, 200 μm (4×), 50 μm (20×). **e** Subcutaneous xenograft tumor formation with RKO cells (shNC vs. shCD36), followed by treatment with intraperitoneal injection of PBS or 3-BrPA (5 mg/kg) or WZB117(5 mg/kg). **f** Representative images of IHC staining of Ki-67, HK2, and GLUT1 on tumor sections. Scale bar, 20 μm (40×). Cell proliferation index was quantified as before. Statistical results are shown as mean ± SEM, *$P < .05$, ***$P < .001$, ****$P < .001$, based on two-way ANOVA or Student's $t$-test. Source data are provided as a Source Data file

## Discussion

CD36 is now gradually presumed to be a metastasis promoter based on its function of fatty acid absorption observed in a broad variety of cancers[3–7], and suppressing CD36 provided good preclinical outcomes in prostate cancer[45]. However, its roles in tumors are far more contentious, even in the same cancer type, CD36 could be either oncogenic or tumor suppressive. As we mentioned before, in glioblastoma, CD36 overexpression in cancer stem cell could promote cancer progression[8], while endothelial CD36 expression played anti-angiogenic and pro-apoptotic functions instead[9,10]. In breast cancer, some articles claimed that lacking CD36 could significantly reduce metastasis[3] and assist the therapeutic effect of tamoxifen[11]. Nevertheless, it was also finely reported that epithelial, endothelial or stromal CD36 expression was negatively correlated with the proliferation and aggressiveness of breast cancer[12–14]. In pancreatic adeno-carcinoma, although it is reported that CD36 on immune cells is indispensable for pancreatic tumor microvesicles to extravasate and form premetastatic foci[16], CD36 may act as a tumor-suppressive gene in pancreatic cancer (PC) as its expression was downregulated in tumors and its deficiency in PC cells predicted large tumor burden and poor prognosis[15], which further suggest the unique cell type-specific, context-specific and function-specific roles of CD36 even in the same cancer type. As the roles of CD36 in CRC remain obscure, we hence sought to delineate the characters of epithelial CD36 in colorectal tumor-igenesis in our study.

In the present work, we found that CD36 was commonly downregulated in human CRC, and revealed a progressive loss of CD36 from colorectal adenomas to carcinomas, which may be due to high methylation levels and polymorphism of CD36 in CRC[46,47]. What is more, CD36 deficiency was related to poor survival and was an unfavorable prognostic indicator of CRC patients. On functional verification, our gain-of-function and loss-of-function experiments in vitro and in vivo clearly suggested an anti-carcinogenic role of CD36 in CRC.

As we know, metabolic reprogramming is a nuclear feature of transformed cells. CD36 has been widely known about its metabolic feature of fatty acid uptake, metastatic cells with CD36 make use of this feature to obtain much energy to invade and survive at distant sites. However, different from metastatic tumors, in locally primary colorectal tumors, we previously confirmed there existed increased fatty acids synthesis but decreased utilization and oxidation of endogenous lipids in human CRC samples[48], CD36 repression may suggest a metabolic protection of cancer cells to take defense to potential lipotoxicity[49]. In addition, tumor cells could utilize the intermediates provided by glycolysis/TCA cycle to biosynthesize NADPH to defend excessive reactive oxygen species (ROS)[50], and it is reported that macrophages from CD36 KO mice have reduced levels of ROS[2]. In this respect, we wondered the different roles of CD36 in primary CRC might result from adaptive metabolic changes by cancer cells to sustain their viability. It is worth noting that fatty acids uptake and oxidation genes, such as CD36, Caveolin-1 and CPT1A, were downregulated in primary tumors but again amplified in metastatic lesions, suggesting a metabolic

shift of CRC cells in metastatic sites[51,52], and these need further study to illustrate the underlying regulatory mechanisms. Aerobic glycolysis and lipid metabolism are closely related in tumors. Although glycolysis has always been deemed as the upstream regulator of lipid metabolism[53], previous findings revealed that CD36-deficient heart cells would present a compensatory mechanism to increase glucose uptake and usage[54,55]. From the perspective of metabolic plasticity, we wondered whether CD36, as a gene involved lipid metabolism, could regulate glucose metabolism in CRC.

Over the past decade, a large amount of evidence has emerged in supporting the critical roles of aerobic glycolysis in promoting tumorigenesis in various cancer types, including CRC[56]. With the renewed interest in glucose metabolism, researchers have realized that increased activity of glycolysis is one of the major con-sequences of certain oncogenic drivers. In CRC initiation and progression, Wnt/β-catenin signaling represents the main path-way involved[57]. c-myc is a pivotal target of β-catenin and reg-ulates thousands of genes[58]. Aberrantly high expression of c-myc is a common basis of tumorigenesis and c-myc is also a key oncogenic driver of glycolysis in normoxia[23]. Here, we demon-strated that CD36 could inhibit glycolysis in a β-catenin activa-tion-dependent c-myc transcriptional way. Ectopic expression of CD36 decreased the expression and nuclear translocation of β-catenin, followed by downregulation of c-myc and downstream glycolytic genes of GLUT1, LDHA, HK2, and PKM2 in CRC cells, which led to decreased glycolytic activity. Pharmacologic inhibi-tions of β-catenin and c-myc abolished CD36-deficiency-mediated glycolytic activation. Impressively, cells with CD36 knockdown were highly addicted to glucose or glycolytic repression from both in vitro and in vivo evidence.

Furthermore, we found that Glypican 4 (GPC4), which belongs to a member of the heparan sulfate proteoglycans (HSPGs) family[59], was a potentially direct interactor of CD36. The co-localization and interaction between CD36 and GPC4 were confirmed and GPC4 protein level was found to be negatively regulated by CD36. Mechanically, CD36 could promote the proteasome-dependent ubiquitination of GPC4 in CRC cells. Although the functional roles of GPC4 in tumors are poorly identified, recent studies have revealed the lipid raft localization of GPC4 is required for the activation of Wnt3a–β-catenin-dependent pathway[41] and GPC4 could maintain the self-renewal of embryonic stem cells in vitro by activating Wnt/β-catenin signaling[60]. In pancreatic cancer, it's reported targeting GPC4 could overcome 5-Fu resistance and cell stemness through sup-pression of Wnt/β-catenin pathway[61]. Herein, GPC4 was repor-ted to be an indispensable downstream effector of CD36 and ectopic GPC4 expression could reverse the tumor suppressive and glycolysis inhibitory functions of CD36 by activating β-catenin/c-myc signaling in CRC cells.

In addition, our report specially constructed recombinant adeno-associated virus (AAV) vector[43] with CD36 knockdown and provided a genetic basis of CD36 disruption in intestinal epithelial cells. It has been reported that Wnt signaling and c-myc protein are important regulators of colitis-associated colorectal tumorigenesis[62]. Our data suggested that AAV-mediated CD36

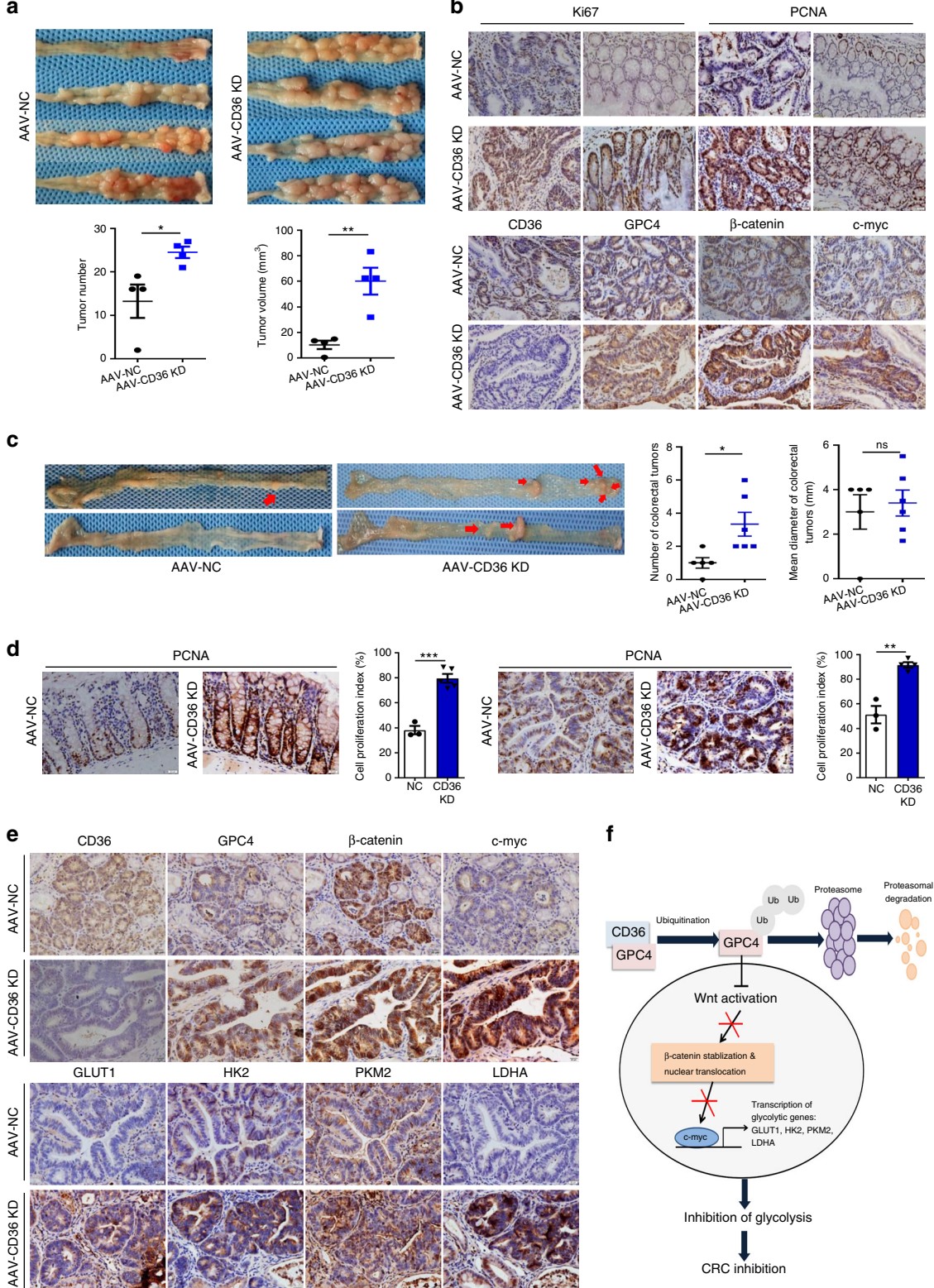

**Fig. 7** AAV-mediated CD36 knockdown promotes CRC. **a** Macroscopic appearance of tumors in the large intestines of AOM-DSS-induced mice. Intestinal tumor numbers and tumor volumes were measured. **b** Representative images of IHC staining of indicated targets on tumor sections. Scale bar, 20 μm (40×). **c** Macroscopic appearance of tumors in the large intestine of $Apc^{Min/+}$ mice, statistical analysis of tumor numbers and sizes in the colon and rectum. **d** Representative IHC staining of PCNA, cell proliferation index was calculated as before. All statistical results are shown as mean ± SEM, based on Student's t-test, *P < .05, **P < .01, ***P < .001. **e** Representative images of IHC staining of CD36, GPC4, β-catenin, c-myc, and downstream glycolytic genes GLUT1, HK2, PKM2 and LDHA on the tumor sections. Scale bar, 20 μm (40×). **f** Schematic diagram summarizing our working model, namely, CD36 can interact with and induce the proteasome-dependent ubiquitination of GPC4, thereby inhibiting β-catenin/c-myc signaling, followed by repressed glycolytic activity and colorectal tumor suppression. Source data are provided as a Source Data file

knockdown promoted more and larger colorectal tumor formation in AOM/DSS-induced CRC. $Apc^{Min/+}$ mice develop adenomatous polyps also due to Wnt pathway activation and c-myc is also essential for $Apc$-mediated intestinal tumorigenesis[63]. In our study, AAV-mediated CD36 knockdown in $Apc^{Min/+}$ mice were found to bear more dysplastic tumors in the colon and rectum. These tumors were characterized by augmented proliferation, elevated expression of GPC4, β-catenin, c-myc and of downstream glycolytic targets.

In conclusion, we found epithelial loss of CD36 expression occurred early in human colorectal adenomas and was extensively absent in primary carcinomas, and decreased CD36 level was strongly associated with malignant properties of CRC cells. Mechanically, the tumor-suppressive effects of CD36 were mediated through promoting the proteasome-dependent ubiquitination of GPC4, and the degraded GPC4 failed to activate β-catenin/c-myc signaling cascades and downstream glycolytic target genes GLUT1, HK2, PKM2 and LDHA (Fig. 7f). Promisingly, as accumulating evidence indicating that regular use of aspirin can significantly improve clinical outcomes of CRC patients[64], and it is reported that the expression of CD36 can be increased in vitro by aspirin treatment[65]. In GEO dataset analysis, treatment of CRC cells obtained from clinical CRC tissues with aspirin and followed microarray data analysis showed CD36 level was upregulated in 87.5% (7 in 8) CRC cells involved (Supplementary Fig. 6f). Collectively, these results strongly argued for CD36 as a colorectal tumor suppressor whose expression may serve as an early biomarker for assessing malignant transformation risk and an intervention target of colorectal neoplasia.

## Methods

**Cell lines**. Normal colon epithelial cell line (FHC) and CRC cell lines (HT29, SW1116, LS174T, CACO2, HCT15, DLD-1, SW480, RKO, HCT116, LoVo, and SW620) were obtained from the US. ATCC with cell authentication. Cells were routinely cultured and maintained in RPMI 1640 (Gibco) supplemented with 10% fetal bovine serum (FBS) and antibiotics (Gibco) according to the ATCC protocols.

**Patient samples**. The Institute Research Medical Ethics Committee of Nanfang Hospital (Guangzhou, China) granted approval for this study. At the time of tissue collection, informed consent was obtained from all patients. Fresh and formalin-fixed tissue samples from patients with colorectal neoplasia were collected randomly from the Department of Gastroenterology or the Department of General Surgery, Nanfang Hospital, affiliated to Southern Medical University, from 1 January 2015 to 1 January 2018. A total of 129 specimens involved 75 patients (11 colonic lesions taken from the Department of Gastroenterology, 54 pair CRC samples and matched normal mucosa and 10 transition zones of CRC samples collected from the Department of General Surgery) were included in our analysis. Fresh samples obtained during endoscopy or surgery were immediately put in formalin or frozen in liquid nitrogen for subsequent experiments. In addition, a tissue microarrays (TMA) derived from 90 patients' resections of CRC and distal normal mucosa were purchased from the National Engineering Center for Biochip at Shanghai. Datasets involved were downloaded from the public GEO (Gene Expression Omnibus) databases (https://www.ncbi.nlm.nih.gov/gds/) and TCGA (The Cancer Genome Atlas) data of COAD (Colon adenocarcinoma) and READ (Rectum adenocarcinoma) were analyzed with GEPIA (http://gepia.cancer-pku.cn/) or UALCAN (http://ualcan.path.uab.edu/index.html) website.

**Mice**. Male athymic nude mice (BALB/c-nu/nu, 4 weeks old) and male BALB/c mice (4 weeks old), purchased from the animal center of Guangdong Province, were used for subcutaneous xenograft models and AOM/DSS-induced CRC mice models, respectively. The $Apc^{Min/+}$ mice were purchased from the GENECHEM Biotech at Shanghai (http://genechem.bioon.com.cn/) and the mouse genotypes were detected with polymerase chain reaction (PCR). All mouse care and experiments were approved by the Institutional Animal Care and Use Committee (IACUC) of Nanfang Hospital. All animal studies were complied with relevant ethical regulations for animal testing and research.

**Protein extraction and western blotting**. Proteins were extracted from tissues and cells and then used for western blot analysis. Cells and tissues were lysed with RIPA buffer containing a protease inhibitor cocktail and a phosphatase inhibitor cocktail (CWBIO). The following primary antibodies were used for western blot analysis: anti-CD36 antibody (1:500, Abcam), anti-β-catenin (1:1000, Proteintech),

anti-cmyc (1:1000, ABclonal), anti-HK2 (1:1000, ABclonal), anti-PKM2 (1:1000, ABclonal), anti-GLUT1 (1:1000, ABclonal), anti-LDHA (1:1000, ABclonal), anti-Cleaved caspase 3 (1:1000, Cell Signaling), anti-Cleaved caspase 9 (1:1000, Cell Signaling), anti-Bcl-XL (1:1000, Cell Signaling), Cell Cycle Regulation Antibody Sampler Kit (1:1000, Cell Signaling), anti-GPC4 (1:500, Abcam), anti-Flag (1:1000, Proteintech), anti-LRP6 (1:1000, Cell Signaling), anti-Phospho-LRP6 (Ser1490) (1:1000, Cell Signaling), anti-GAPDH (1:1000, Proteintech), anti-Lamin B (1:1000, ABclonal), anti-GFP (1:1000, Proteintech). Detailed antibody information was listed in Supplementary Table 3. All uncropped scans for blots were presented in corresponding Source Data file.

**Quantitative real-time PCR**. Total RNAs from cells or tissues were extracted with TRIzol reagent (TaKaRa) in accordance with the manufacturer's instructions. cDNAs were generated with PrimeScript RT-PCR Kit (TaKaRa). The expressions of mRNA were analyzed using SYBR Premix Ex Taq (TaKaRa) with a LightCycler 96 Detection System (Roche) using GAPDH or β-actin for normalization. The relative quantification of qRT-PCR was calculated according to the work by Pfaffl[66]. Primers used were detailed in Supplementary Table 4.

**Immunohistochemistry**. IHC staining of paraffin-embedded human or mice tumor sections were performed according to standard protocols. Sections were deparaffinized, rehydrated, subjected to antigen retrieval, and blocked with 3% hydrogen dioxide and goat serum, followed by incubating in primary antibodies overnight at 4 °C. First antibodies used were listed as follows: anti-CD36 antibody (1:200, Abcam), anti-β-catenin (1:200, Genetex), anti-cmyc (1:200, GeneTex), anti-HK2 (1:500, ABclonal), anti-PKM2 (1:500, ABclonal), anti-GLUT1 (1:250, Abcam), anti-LDHA (1:250, GeneTex), anti-GPC4 (1:200, Proteintech), anti-GFP (1:500, Proteintech), anti-Ki-67 (for mouse, 1:800, Cell Signaling), anti-Ki-67 (for human, 1:800, Cell Signaling), anti-PCNA (1:1000, Proteintech). Next day, the sections were put in room temperature for 30 min to rewarm, followed by secondary antibody incubation for 1 h in room temperature and DAB staining was performed with IHC assay kit (Maixin, China). Counterstaining was carried with hematoxylin for 2 min. Images were taken with OLYMPUS DP22 microscope. Antibodies were shown in Supplementary Table 3.

**Stable cell line generation and plasmids transfection**. Lentiviral vectors plasmids were constructed by GENECHEM Biotech at Shanghai, China (http://genechem.bioon.com.cn/). Flag-tagged CD36-overexpressed vectors (LV-CD36) and control vectors (LV-RFP) were transfected into SW480 and LoVo cells, while CD36 shRNA (shCD36) and control short hairpin RNA (shNC) were transfected into CACO2 and RKO cells to generate cells with stable knockdown of CD36. Transfection procedures were performed according to provided protocols. Briefly, cells ($1 \times 10^4$ per well) grown in 24-well plates were transduced with lentiviruses for 24 h, then cells were selected with puromycin (4 μg/ml) 48 h post transduction for 6–10 days and expanded. Overexpression or knockdown of CD36 was assessed by qRT-PCR and western blot. Transfection of SW480 and LoVo cells (LV-CD36 and LV-RFP) with HA-GPC4 or Empty vector was performed using Lipofectamine 3000 (Life Technologies) with 10 μg plasmid DNA. Cells were incubated for 48 or 72 h after transfection before testing for transgene expression or performing downstream experiments. The primers sequences were listed on Supplementary Table 4.

**CCK8, colony formation, and MTT assay**. For cell proliferation assays, cells (1000 per well) were cultivated on 96-well plates and cell proliferation were detected for 6 days with cell counting kit-8 (DOJINDO Laboratories) at 450 nm. For the colony formation assays, cells (500 cells per well) with different treatment were cultivated in 6-well plates. At the end of experiments, colonies formed were washed with phosphate buffer (PBS), fixed in methanol and stained with 0.1% crystal violet. Colonies containing more than 50 cells for each well were counted (in triplicate). For MTT assays, cells (1000 per well) were cultivated on 96-well plates and cultured for 24 h, then medium was refreshed with RPMI 1640 containing 0.5 mM glucose, 20 μM Cytochalasin B (MCE), or 10 μg/ml 3-BrPA (Selleck) for 48 h. Then cell proliferation was detected with MTT (Beyotime, China) at 570 nm.

**Migration and invasion assay**. To access cell migration, $1.0 \times 10^5$ cells were seeded into the 8-μm-pore upper chambers in serum-free RPMI1640 and incubated in RPMI1640 with 10% FBS of the lower chamber of 12-well plates (Corning Star). After normal culture for 12–48 h, cells were permeabilized with 100% methanol and then stained with 0.1% crystal violet. For invasion assay, 30 times dilution of matrigel (BD Biosciences) were used to coat the upper chambers and $1.0 \times 10^6$ cells were seeded into the chambers, and other procedures were performed as in the migration assay. Images were taken with OLYMPUS DP22 microscope and cells were quantified in at least five random microscopic fields.

**Glucose uptake assay**. 2-NBDG (Life Technologies) was used as a glucose tracer. Briefly, $1 \times 10^5$ cells per well were seeded in 6-well plates in quadruplicate and incubated overnight at 37 °C with 5% $CO_2$. Next day, cells were starved for glucose for 4 h, then one well was incubated with RPMI1640 medium with 25 μM glucose

as a negative control, and the remaining three wells were cultivated with 25 μM 2-NBDG for 2 h. After incubation, cells were digested and washed twice with PBS. The mean fluorescence intensity of cells was measured by flow cytometry with excitation light at 488 nm.

**Lactate assay**. To measure lactate production, $1 \times 10^5$ cells per well were seeded in 24-well plates in triplicate for 24 h, then the medium was refreshed with RPMI1640 containing 1 mM glucose overnight. The next day, culture medium was harvested and lactate concentration was detected and measured according to commercial assay kits (Abcam).

**ATP assay**. To measure ATP production, $1 \times 10^4$ cells per well were seeded in 96-well plates in quintuplet for 24 h and then the medium was refreshed with 1 mM glucosec overnight. Relative ATP concentration was measured according to commercial assay kits (Abcam).

**Flow cytometry of apoptosis and cell cycle**. For apoptosis analysis, stable transfected cells under detection were collected after normal condition, low glucose (0.5 mM) or 5-Fu (50 μg/ml) treatment for 24–48 h, respectively. Cells were then collected and apoptosis was then detected with Annexin V-FITC/DAPI cell apoptosis detection kit. For cell cycle analysis, adherent cells under detection were synchronized and starved in RPMI1640 without FBS for 12 h, then the medium was refreshed with RPMI 1640 supplemented with 10% FBS and cells were cultured for another 24 h, then cells were collected and fixed with 70% ethanol for 24 h and then resuspended in 1 ml of PBS containing RNase and DAPI. The detailed procedures were applied according to instructions provided (keyGen Biotech).

**Co-IP assay**. Total proteins were extracted with cell lysis buffer supplemented with protease inhibitor and phosphatase inhibitor. Lysate (100 μg protein) was incubated with anti-CD36 (1:50, Santa Cruz), anti-GPC4 (1:50, Abcam), anti-Flag (1:50, Proteintech), anti-FK2 (1:100, EMD Millipore) or IgG (as a negative control, 1:1000, Cell Signaling) at 4 °C overnight. Then the protein–antibody complex was incubated with protein A/G magnetic beads for 5 h at 4 °C. Immunoprecipitation was then collected by centrifugation at $1000 \times g$ for 5 min at 4 °C and washed the beads complex four times with PBS. After final wash, protein A/G magnetic beads eluted by boiling in 5× SDS sample buffer before western blot. Antibodies and dilutions used were detailed in Supplementary Table 3.

**Silver staining**. Silver staining was performed according to the protocol provided by Beyotime Technology. Briefly, the gels were fixed in 50% ethanol/10% acetic acid for 40 min after electrophoresis, washed in 30% ethanol for 10 min and in Milli-Q water for another 10 min. The gels were incubated with silver staining sensitizer for 2 min and then with silver nitrate for 10 min. Afterwards, the gels were put in Milli-Q water for 1 min, removed, and in the developing solution. When clear staining was achieved, the gels were transferred to stop solution for 10 min. The stained gels were stored in Milli-Q water at 4 °C.

**Chromatin immunoprecipitation (ChIP) assay**. ChIP experiments were performed according to the protocol of Chromatin Immunoprecipitation kit (Chromatrap). Immunoprecipitation reactions were performed with 5 μg antibodies against β-catenin (1:20, GeneTex) or with IgG used as a negative control. Purified DNA was then suspended for following qRT-PCR analysis using primers of MYC or Cyclin D1 promoters. PCR products were then run on 1.5% agarose gels and visualized with ethidium bromide. Relative chromatin enrichment was calculated as the amount of amplified DNA normalized to input and relative to values obtained after normal IgG immunoprecipitation.

**Ubiquitination assay**. Cells were incubated in culture media and total proteins were extracted and Co-IP was performed using anti-GPC4 or anti-IgG, respectively. The protein complexes were then subjected to western blot using anti-K48-linked poly-ubiquitination antibody (1:2000, Abcam) and anti-FK2 antibody (1:2000, EMD Millipore) to evaluate the proteasome-dependent ubiquitination level.

**Immunofluorescence**. Cells were cultured in the confocal dish for 48–72 h and then washed with PBS for three times and fixed in 4% paraformaldehyde for 30 min, permeabilized with 0.5% Triton X-100, followed by blocking with goat serum, then cells were subjected to staining with anti-CD36 antibody (1:50, Santa Cruz), anti-β-catenin (1:100, Santa Cruz), anti-cmyc (1:100, GeneTex) or anti-GPC4 (1:100, Proteintech) antibody, followed by Alexa-488-conjugated goat anti-mouse secondary antibody (1:500, Bioss) or Alexa-647-conjugated goat anti-rabbit secondary antibody (1:500, Bioss) for imaging. The cells were then counterstained with 4′,6-diamidino-2-phenylindole (DAPI) as a nuclear indicator and imaged with a confocal laser-scanning microscope (Carl Zeiss, LSM 880 with Airyscan).

**Subcutaneous xenograft model**. Briefly, stable transfection cells $(1.0 \times 10^7)$ were subcutaneously implanted into the bilateral flanks of 4-week-old nude mice. In the

drug delivery experiments, ten days after subcutaneous injection and palpable tumors were present in all animals, the mice were then randomly divided into 3 groups, and 3-BrPA (Selleck) or WZB117 (Selleck) were then injected intraperitoneally every 2 days at the concentration of 5 mg/kg/d, and PBS as a control. The tumor growth was monitored and tumor volumes (mm$^3$) were estimated as follows: tumor volume = $L \times W^2/2$, where $L$ is the length and $W$ is the width. Tumors were harvested when reached a diameter about 2.0 cm. Tumors were excised and rapidly put in formalin for further IHC analysis.

**Intrasplenic inoculation model**. CD36-overexpressed or control SW480 cells were injected at a final concentration of $1 \times 10^6$ cells/50 μl PBS into spleens of each 4-week-old male Balb/c nude mice (7/group). Mice were sacrificed after 4 weeks, and the spleens and livers were dissected, and the number of liver metastasis was recorded. Then the tissues were fixed in 10% neutral-buffered formalin for following hematoxylin and eosin staining and microscopic analysis.

**AAV9-GFP construction and tail vein injection**. CD36 knockdown and control recombinant Adeno-associated virus-green fluorescence protein vectors 9 (AAV9-GFP) were constructed (GENECHEM Biotech). The administration procedures were performed according to previous studies[44]. Briefly, $5 \times 10^{10}$ physical particles of AAV in 200 μl of PBS were injected into the tail veins of 4-week old BALB/c and $Apc^{Min/+}$ mice. Small intestine and colon were removed for GFP expression evaluation with western blots and IHC.

**AOM/DSS-induced mice model**. Detailed induction process was shown in Supplementary Fig. 5d. For generation of the chemically induced CRC model, firstly, AAVs were injected into the tail veins of 4-week-old BALB/c mice, 2 months later, mice were then intraperitoneally injected with 10 mg/kg body weight azoxymethane (AOM; Sigma-Aldrich). Next day, 2.5% dextran sulfate sodium (DSS; MW 40,000–50,000 Da, MP) was dissolved in distilled water and administered for seven days, followed by 14 days of normal drinking water. This cycle was repeated three times. One week later, the mice were sacrificed and the large intestines were dissected for recording tumor load, tumor size and intestinal weight. Then the intestinal tissues were frozen in liquid nitrogen or fixed in 10% neutral-buffered formalin for following molecular biological investigations[67].

**Statistics**. The results were shown as mean ± SEM. Survivals related to CD36 expression were evaluated by the Kaplan–Meier survival analysis. A two-tailed, unpaired, or paired Student $t$ test was used to compare the variables of two groups, and one-way or two-way ANOVA were performed for multi-group comparisons. Co-localization of IF[68] and the analysis of mean of interest of domain (IOD) in IHC staining were calculated with Image Pro Plus. Significant differences were reflected with $*P < 0.05$, $**P < 0.01$, $***P < 0.001$, and $****P < 0.0001$. Statistical details are included in the respective figure legends.

**Reporting Summary**. Further information on research design is available in the Nature Research Reporting Summary linked to this article.

## Data availability

The Gene Expression Omnibus (GEO) data and TCGA data referenced during the study are available in a public repository from the GEO website (https://www.ncbi.nlm.nih.gov/geo/), GEPIA website (http://gepia.cancer-pku.cn) and UALCAN website (http://ualcan.path.uab.edu/cgi-bin/ualcan-res.pl). The authors declare that all the other data supporting the findings of this study are available within the article and its Supplementary Information files and from the corresponding author on reasonable request. The source data underlying Figs. 1b-d, 1f, 2a-f, 3a-d, 3f, 4a-f, 5a-f, 6a-f, 7a, 7c, 7d and Supplementary Figs. 1c, 2a, 2c–e, 4d, 5a-b, 5e, 6b, 6e, 7b are provided as a Source Data file.

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

## Acknowledgements

The study was substantially sponsored by the National Natural Science Foundation of China [81672992, 31601023, 81872470]; Guangdong Natural Science Funds for Distinguished Young Scholar [2015A030306015]; Pearl River Nova Program of Guangzhou, Guangdong Province [2014J2200015]; Excellent Young Teachers Program of Higher Education of Guangdong Province [YQ2015036]; Guangdong Program for Support of Top-notch Young Professionals [2015TQ01R279]; the Foundation of President of the Nanfang Hospital [2015C020]; the Natural Science Foundation of Guangdong Province, China [2016A030310387, 2017A030313154, 2017A030313588, 2018A030313547]; Medical Scientific Research Foundation of Guangdong Province, China [A2016237]; the Science and Technology Planning Project of Guangzhou, China [201707010214].

## Author contributions

Study concept and design: Y.F., Z.-Y.S., D.-H.W., Y.D.; development of methodology: Y.F., Z.S., D.-H.W., Y.D.; draft of the manuscript and data acquisition: Y.F., Z.S., Y.Z., X.F., K.C., Y.-S.-L., S.-M.P.; analysis and interpretation of data: Y.F., Z.-Y.S., Y.-Z.Z., D.-H.W., Y.D.; administrative, technical, or material support: H.-J.D., D.-H.W., Y.D.; study funding and supervision: Z.-Y.S., D.-H.W., Y.D.

## Additional information

**Competing interests:** The authors declare no competing interests.

