## [Peer Review File · Nature Communications]

Reviewers' Comments:

Reviewer #1:

Remarks to the Author:

Fang and colleagues report here that loss or reduced expression of CD36 in colorectal cancer is associated with poor patient survival. They demonstrated that knockdown of CD36 in mouse colon resulted in increased number of colon tumors in both AOM-DSS-induced and Apc min models. Mechanistically, their data suggest that loss of CD36 leads to accumulation of GPC4 protein, which in turn stabilizes β -catenin and up-regulates its target c-MYC. Subsequently, c-MYC turns on the expression of proteins involved in glycolytic pathway including GLUT1, HK2, PKM2 and LDHA. A large amount of data are presented in this manuscript, which supports the conclusions made by the authors. However, there are some concerns need to be addressed.

1. The proposed mechanism seems to be paradoxical to the current paradigm of colorectal tumorigenesis. It is well documented that most colorectal cancers have mutations in either APC or β -catenin, thereby stabilizing β -catenin and activating its transcriptional activity. Numerous studies have demonstrated that loss of Apc or activation of β -catenin is sufficient to drive the initiation of colorectal tumorigenesis. In this regard, why do colorectal cancers need both loss of CD36 and mutations in APC or β -catenin (e. g. in cell lines HT29, SW116, HT15 and HCT116 shown in Fig. S2a)?
2. Figure 1a shows that CD36 expression levels vary across different datasets. Were different RNA profiling platforms used for these studies? If so, the author should not plot all the data sets in one graph.
3. In Figure 1 f & g, what does each of the curves indicate?
4. How does GPC4 regulate β -catenin protein stability?
5. Does 3-BrPA or Cyto-B inhibits xenograft tumor growth of RKO cells with CD36 knockdown?
6. Remove the Chinese characters in Fig. S1d.

Reviewer #2:

Remarks to the Author:

In this manuscript, the authors discover the critical role of CD36 during colorectal cancer progress. They uncover that CD36 acts as a tumor suppressor by interacting, promoting the degradation of GPC4, which is a regulator for b-catenin/cMyc dependent expression of genes in the glycolysis pathway. This study critically links lipid and glucose metabolism in colorectal cancer. However, specific questions concerning about the robustness of data should be addressed before publication.

1. In figure 1, the authors show that the mRNA and protein level of 36 is lower in colorectal cancer than paired normal tissue. As CD36 is a cell surface receptor, how about the level of CD36 on the cell surface in those samples? Is that higher in tumor than normal tissue?
2. On figure 2b, it is not logical for the authors to examine the apoptosis in control and CD36 knockdown since there should not be much apoptosis in control cells. The author would rather compare apoptosis between control and CD36 overexpressed cell line and/or compare control and CD36 knockdown cells under stressed condition, like nutrient/growth factor starvation or chemo drug treatment.
3. In consistent with apoptosis data, the author should also show CD36 overexpressed cells in colony forming assay under glucose deprivation condition.
4. A panel of control cells treated with b-catenin and myc inhibitors should be compared side by

side with the data in figure 3d. This is critically related to why CD36 knockdown cells are more sensitive compared to control cells in fig 3e.

5. To be physiological relevant of CD36 loss in tumor, the stability assay for GPC4 is only necessarily to be performed in CD36 knockdown cells in fig 4d, but such assay should be done in at least two cell lines, triplicated and quantified with error bar. It would also be curious that whether GPC4 has a high turnover rate in CD36 high cell lines and whether CD36 knockdown is sufficient to block its turnover. This could be done by treating those cells with MG132 as performed in fig 4e.

6. If regular IP process is used for detecting the endogenous protein ubiquitination, a reciprocal way by pulling down ubiquitinated proteins (by FK2 antibody) followed by blotting for substrate protein is critical to show the assay specificity. Molecular weight marker should be included for any blots showing protein ubiquitination.

7. It is also critical to determine whether the ubiquitination of GPC4 is K48 linked.

8. How does GPC4 regulates b-catenin in the colorectal cancer cell line. Is it via the same mechanism described previously? The author should briefly test this.

Minor points

1. On figure S1d, every labeling should be English.

2. The author should show FDR value for the GSEA analysis in figure 2d.

3. Statistical data should be presented for the immunofluorescence assay in fig 3c.

4. Input blots for SW480 is shifted in fig 4f.

5. The authors should explain why MG132 treatment in control cells does not increase the b-catenin level in figure 5a, since b-catenin is also a protein with high turnover rate. Lysosome inhibitor should be used as a control for MG132.

Reviewer #3:

Remarks to the Author:

In this work, Fang and colleagues show that the expression of CD36 is reduced in colorectal cancer samples and during the progression from adenomas to carcinomas. They show that CD36 acts as a suppressor of primary tumor proliferation through the inhibition of aerobic glycolysis.

Mechanistically this happens through the regulation of the stability of the protein GPC4, which in turn modulates the activity of Wnt signaling. Thus, high expression of CD36 results in a higher rate of degradation of GPC4 which in turn downregulates beta-catenin and Myc signaling (and the glycolytic downstream targets of this pathway).

This is an interesting work that provides novel insight into the role of CD36 in primary tumor growth. CD36 has recently attracted much attention as a critical mediator of the metastatic potential of metastatic stem cells. However, its role in the biology of primary tumors has not been well characterized so far. The work is therefore potentially interesting to the readership of Nat Communications. However, there are certain issues that in my opinion need to be addressed before the paper can be recommended for publication.

Main comments:

1) In the introduction, the authors nicely claim that CD36 is essential for metastasis (or tumor progression) of glioblastoma, oral cancer, and hepatocellular cancer. However, in Pascual et al, it was also shown to be essential for metastasis in melanoma and luminal breast cancer. In addition, solid reports have shown that CD36 is essential for metastasis of serous ovarian cancer (Lengyel group), melanoma (White group at Memorial), pancreatic adenocarcinoma (a recent report showing the metastasis-promoting exosomes depend on CD36), and cervical cancer. These works should be cited and commented in the introduction and discussion.

2) Minor comment: in Figure 2e the KD of CD36 in Caco2 cells seems to reduce proliferation rather than increase it as claimed in the text. I think this is a simple mislabelling of the figure.

3) Why are Caco2 and RKO cells so apoptotic in culture? A 20% of apoptosis is an anomaly and suggests that the culture conditions are suboptimal. Is there an explanation for this?

4) Major point: the authors nicely show that cells that overexpress CD36 are very resistant to glucose deprivation. I think this is a very interesting result with possibly broad consequences. As primary tumors grow in size it is well known that they become hypoxic which generates areas of low glucose levels. There is substantial literature showing that hypoxic areas favor or promote metastatic spreading of tumors (whilst showing low proliferative rates of tumor cells within these areas). It should be noted that CD36+ metastatic stem cells were identified precisely as long-term quiescent cells in the primary tumor. Hence, CD36 would not favor the proliferation of primary tumor cells but would favor the metastatic spreading of cells. All of this would be consistent with the findings shown in this paper. However, the authors have not explored the effect of CD36 in CRC metastasis. I think this is quite relevant considering that a nice report by Nath and Chan has shown that high expression of CD36 correlates with metastasis in CRC patients.

I, therefore, think that the paper would significantly gain in relevance if the authors could study the possible dual role of CD36 in primary tumors versus metastasis. The results shown by the authors regarding primary tumor growth are solid. Yet I think it is necessary for the authors to study what is the role of CD36 in the metastatic potential of the CRC cell lines they are using in this study. That is, does the overexpression of KD of CD36 affect the metastatic potential of CRC cells? This should be tested by orthotopic inoculations, intrasplenic inoculation (which should mainly result in liver metastases), and intravenous inoculation (which should primarily result in lung metastases). Providing a potential dual role of CD36 in primary tumors versus metastases would be a very interesting finding.

5) Quantification and statistics should be provided for the results shown in Figure 3c.

6) In line 238 it should say Figure 3f instead of Figure 5f.

7) The Western blots results of GPC4 in Figure 4a are rather weak. I am not doubting at all at the validity of the conclusions, but I think the authors should show clearer blots, or at least some quantification and statistics.

8) The results shown in Figure 4b are likewise unclear. The reported colocalization of CD36 and GPC4 is hardly visible. Much clearer IFs should be provided and a detailed image analysis performed to demonstrate that both proteins indeed colocalize.

9) Results shown in Figure 6a and 6b are consistent with a role for CD36 in promoting quiescence. This comment is related to comment 4. Again, I think it is very important that the authors study the potential role of Cd36 in CRC metastasis considering the increasing literature supporting a role for this protein in metastasis. It might very well be that CD36 is also metastasis suppressive in CRC and this would indeed be an interesting finding. But whichever the results I think it is very important that the authors perform the intrasplenic, orthotopic and intravenous inoculations to test if modulating the expression of CD36 has any impact over the metastatic potential of CRC cells.

10) Can the authors rule out that the effect of the AAVs modulating the expression of CD36 is due to non-epithelial cells? Is the expression of CD36 affected in the colorectal stroma of mice infected with the adenovirus? I think this is a relevant issue considering the recent report in which deletion of CD36 has been shown to cause intestinal inflammation, but that the phenotype is primarily due to loss of CD36 in the endothelium rather than the epithelium (Cifarelli V et al., 2017).

11) A recent report has shown that CD36 affects insulin signalling and glucose metabolism (Samovski D et al., Diabetes 2018). This paper should be acknowledged and discussed in the discussion section when the authors comment on the role of CD36 in glycolysis.

12) The authors might also want to discuss in their paper recent papers showing that polymorphisms in CD36 affect the risk of developing colorectal cancer.

As the last point, I would like to apologize to the authors for having taken so long to submit my review of their work. As an author myself, I know how stressful this can be. This has been due to personal reasons.

Dear Editors and Reviewers :

Thank you for your kind letter regarding our manuscript entitled “*CD36 inhibits β -catenin/c-myc-mediated glycolysis through ubiquitination of GPC4 to repress colorectal tumorigenesis*” (NCOMMS-18-24821A). We sincerely appreciate the Reviewer’s suggestions and these comments are all valuable and helpful for revising and improving our paper, as well as the important guiding significance to our researchers. According to the associate editor and reviewer’s comments, we have made extensive modifications to our manuscript and the point-by-point responses are listed below this letter. Changes to our manuscript were all outlined in red in the paper and we sincerely hope to have an opportunity to publish this paper in *Nature Communications*.

Replies to Reviewer #1 comments

1. The proposed mechanism seems to be paradoxical to the current paradigm of colorectal tumorigenesis. It is well documented that most colorectal cancers have mutations in either APC or β -catenin, thereby stabilizing β -catenin and activating its transcriptional activity. Numerous studies have demonstrated that loss of Apc or activation of β -catenin is sufficient to drive the initiation of colorectal tumorigenesis. In this regard, why do colorectal cancers need both loss of CD36 and mutations in APC or β -catenin (e. g. in cell lines HT29, SW116, HT15 and HCT116 shown in Fig. S2a)?

Response: Thanks very much for your valuable comments. As the Reviewer mentioned, human germ-line mutations in the adenomatous polyposis coli (APC) gene always result in familial adenomatous polyposis, and up to 80% of CRCs show mutations in both adenomatous polyposis coli alleles. Canonical WNT signaling is crucial in both normal development and carcinogenesis, the key event of WNT signaling is the activation of β -catenin, which is crucial for colorectal carcinogenesis, and the activation of WNT signaling is a consequence of APC loss¹. In our paper, we found that there had a progressive loss in CD36 expression as adenomas became larger and more dysplastic and CD36 staining in the epithelial component of carcinomas was almost absent, as shown in Fig. 1d and Supplementary Fig. 1c, so we speculated that CD36 loss might be an early event in CRC development and associated with the malignant transformation of adenomas. However, these data could not suggest CD36 causes malignant transformation of normal colon

mucosal cells just like APC mutation or activated β -catenin, maybe CD36 loss is a result event from the malignant transformation. Secondly, we found that CD36 could interact with GPC4 and promote the ubiquitination of GPC4 in CRC cells, and GPC4 could be an activator of β -catenin/c-myc signaling in CRC cells, in this regard, we think that loss of CD36 probably could partly maintain the constitutive activation of β -catenin and drive the development of colorectal tumorigenesis unlike the regulative mechanism of APC mutation.

2. Figure 1a shows that CD36 expression levels vary across different datasets. Were different RNA profiling platforms used for these studies? If so, the author should not plot all the data sets in one graph.

Response: Thank you very much for your kindly suggestions. The public data used were from different RNA profiling platforms, and the comparison has been separately presented in Fig.1a.

3. In Figure 1 f & g, what does each of the curves indicate?

Response: In Figure 1 f & g, the solid line represented survival curve and the two dashed lines indicated the Confidence interval of 0.95. And only survival curves were present in our revised vision.

4. How does GPC4 regulate β -catenin protein stability?

Response: Considering the Reviewer's comments, we transfected HEK293T cells with control or GPC4-overexpressed plasmids to generate GPC4-overexpression cell lines. After 96 hours of transfection, we treated cells with different concentrations of Wnt3a for 1 hour and then collected the cells for following qRT-PCR and Western blot analysis. As previous work has claimed that GPC4 could activate Wnt3a- β -catenin pathway via promoting phosphorylation of LRP6 at Ser1490², so we analyzed the mRNA and protein level of β -catenin as well as the protein expression of LRP6 at Ser1490 after Wnt3a treatment. As shown in Supplementary Fig. 5e, we found Wnt3a treatment could result in a gradient increase of β -catenin mRNA, and the upregulation was significantly higher in GPC4-overexpressed cells than were in control cells. Western blot analysis further identified a higher phosphorylation of LRP6 at Ser1490 and β -catenin expression after Wnt3a treatment in GPC4-overexpressed 293T cells.

5. Does 3-BrPA or Cyto-B inhibits xenograft tumor growth of RKO cells with CD36 knockdown?

Response: Thanks for your valuable comments, we have conducted additional animal experiments accordingly. We used 3-BrPA (a HK2 inhibitor) and WZB117(a GLUT1 inhibitor for intraperitoneal injection)³ to test the inhibitory

effect on xenograft tumor growth of RKO cells with CD36 knockdown. Results revealed that treatment of mice with 3-BrPA or WZB117 could indeed lead to sharp growth inhibitions on CD36-repressed RKO tumors. A significant decrease in tumor volume was observed in shCD36+3-BrPA (76.7% reduction) and shCD36+WZB117 (69.8% reduction) group as compared to the shCD36+PBS group, as shown in Fig.6c. However, drug delivery seemed to induce a significant growth promotion of the control tumors, we think it might result from a more supportive environment as drugs efficiently suppressed the tumor growth in the opposite side, or maybe there involved other complicated mechanisms we were not able to figure them out yet. Growth regression was further confirmed by quantifying with Ki67 staining, and HK2 and GLUT1 protein expressions were also analyzed in Fig.6d.

6. Remove the Chinese characters in Fig. S1d.

Response: We are very sorry and we have corrected it in the revised manuscript.

Thank you very much for your kind comments and suggestions.

Replies to Reviewer #2 comments

1. In figure 1, the authors show that the mRNA and protein level of 36 is lower in colorectal cancer than paired normal tissue. As CD36 is a cell surface receptor, how about the level of CD36 on the cell surface in those samples? Is that higher in tumor than normal tissue?

Response: Thanks for your nice questions. Although our measurement of the mRNA and protein level of CD36 in human tissues could not show the accurate localization of CD36, I hope our IHC results could answer your questions. In our IHC staining of human CRC tissues, we found that CD36 was mainly located on the cell membrane and cytoplasm of the epithelial component of normal mucosa, while its staining in the epithelial component of carcinomas were almost absent, and it was rather rare to observe CD36 staining in the stroma of both normal and cancerous tissues. In the tissue microarray (TMA) consisting of 90 pairs of human colon cancer samples, we also detected the same expression pattern. Taking these data into consideration, although we could not further extract the cell membrane of CRC tissues due to a lack of fresh tissues, we think it may be logical to reason that the difference of the mRNA and protein expression of CD36 in CRC tissues were mainly due to the discrepancy of membrane and cytoplasmic CD36

expression in epithelial cells.

2. On figure 2b, it is not logical for the authors to examine the apoptosis in control and CD36 knockdown since there should not be much apoptosis in control cells.

The author would rather compare apoptosis between control and CD36 overexpressed cell line and/or compare control and CD36 knockdown cells under stressed condition, like nutrient/growth factor starvation or chemo drug treatment.

Response: Thank you so much for this question. We are very sorry for our neglect of clear indication that the apoptosis showed were under 5-fluorouracil (5-Fu) treatment. Considering your kind suggestion, we have conducted the experiments more cautiously to compare apoptosis between control and CD36 overexpressed cell lines (SW480 and LoVo) and between shNC and shCD36 cells (RKO and CACO2) under normal and 5-Fu treated conditions, respectively. All experiments were repeated three times. Protein expression of apoptosis markers (cleaved caspase-3, cleaved caspase-9, Bcl-XL) were also detected with Western blot analysis. Results showed LoVo cells displayed a higher apoptosis rate after CD36 overexpression and CD36 knockdown led to a lower apoptosis rate in CACO2 cells under normal conditions, respectively. However, when treating cells with 5-Fu, overexpression of CD36 induced a significant increase of apoptosis as compared with their negative controls in both SW480 and LoVo cells, and both RKO and CACO2 cells with shCD36 showed sharply less apoptosis than their controls. Protein expression of apoptosis markers revealed consistent changes. Related pictures and statistics have been presented in Fig.2b and Supplementary Fig.2c.

3. In consistent with apoptosis data, the author should also show CD36 overexpressed cells in colony forming assay under glucose deprivation condition.

Response: As suggested by the Reviewer, we performed additional experiments to detect the colony forming ability of CD36 overexpressed cells under glucose deprivation condition. As shown in Fig.3a and Supplementary Fig.3b, SW480 and LoVo cells with CD36-overexpression showed significantly less glucose addiction than their negative controls, and gradient deprivation of glucose significantly inhibited the viability of control cells.

4. A panel of control cells treated with b-catenin and myc inhibitors should be compared side by side with the data in figure 3d. This is critically related to why CD36 knockdown cells are more sensitive compared to control cells in fig 3e.

Response: We fully agree with the Reviewer's comment and performed additional Western blot analysis to support our conclusions. As expected, CRC cells with CD36 knockdown were far more sensitive to both β -catenin inhibitor XAV-939 and c-myc inhibitor 10058-F4, as shown by more severe repression of the protein levels of downstream glycolytic genes HK2, PKM2, LDHA, GLUT1 in RKO and CACO2 cells with shCD36 than in control cells. Related pictures and statistics have been presented in Fig.3d and Supplementary Fig.3e.

5. To be physiological relevant of CD36 loss in tumor, the stability assay for GPC4 is only necessarily to be performed in CD36 knockdown cells in fig 4d, but such assay should be done in at least two cell lines, triplicated and quantified with error bar. It would also be curious that whether GPC4 has a high turnover rate in CD36 high cell lines and whether CD36 knockdown is sufficient to block its turnover. This could be done by treating those cells with MG132 as performed in fig 4e.

Response: Considering the Reviewer's suggestions, we performed Western blot analysis in RKO and CACO2 cells with shCD36 or shNC to test the stability of GPC4 after CHX treatment and repeated for three independent times. Blotting and statistical results showed in Fig.4d indicated that the half-life periods of GPC4 were much longer in RKO and CACO2 cells with shCD36 than were in shNC cells. Moreover, we also treated above-mentioned cells with MG132, and results revealed that MG132 treatment could significantly increase the GPC4 protein expression in control cells, while only a slight upregulation of GPC4 was observed in cells with CD36 knockdown. The blotting pictures and relative protein expression were present in Fig.4e. These data strengthened that CD36 knockdown is sufficient to block the degradation of GPC4.

6. If regular IP process is used for detecting the endogenous protein ubiquitination, a reciprocal way by pulling down ubiquitinated proteins (by FK2 antibody) followed by blotting for substrate protein is critical to show the assay specificity. Molecular weight marker should be included for any blots showing protein ubiquitination.

Response: As suggested, we performed Co-IP with total ubiquitin FK2 antibody to examine the endogenous ubiquitination of GPC4 and a reciprocal way by pulling down FK2 with anti-GPC4 antibody. As shown in Fig.4f, results confirmed the interaction between FK2 and GPC4 in both RKO and CACO2

cells, and CD36 overexpression in SW480 and LoVo cells could significantly increase the abundance of FK2 co-precipitated by GPC4.

7. It is also critical to determine whether the ubiquitination of GPC4 is K48 linked.

Response: Thanks very much for your valuable suggestions. As K48-linked polyubiquitination generally targets proteins for proteasomal degradation⁴, so identifying whether ubiquitination of GPC4 is K48-linked is necessary and will make our results more convincing. We conducted Co-IP with anti-GPC4 antibody and performed Western blot with Anti-Ubiquitin (linkage-specific K48) antibody accordingly, results showed overexpression of CD36 could significantly increase K48-linked polyubiquitination of GPC4 in both SW480 and LoVo cells. Related pictures were presented in Fig.4f.

8. How does GPC4 regulates b-catenin in the colorectal cancer cell line. Is it via the same mechanism described previously? The author should briefly test this.

Response: We agree with the Reviewer's comments and we then transfected HEK293T cells with control or GPC4-overexpressed plasmids to generate GPC4-overexpressed cell lines. After 96 hours of transfection, we treated cells with different concentrations of Wnt3a for 1 hour and then collected the cells for following qRT-PCR and Western blot analysis. As previous work has claimed that GPC4 could activate Wnt3a- β -catenin pathway via promoting phosphorylation of LRP6 at Ser1490, so we analyzed the mRNA and protein level of β -catenin as well as the protein expression of LRP6 at Ser1490 after Wnt3a treatment. As shown in Supplementary Fig. 5e, Wnt3a treatment could result in a gradient increase of the mRNA of β -catenin, and the upregulation was significantly higher in GPC4-overexpressed cells than were in control cells. Western blot analysis further identified a higher phosphorylation of LRP6 at Ser1490 and β -catenin expression after Wnt3a treatment in GPC4-overexpressed 293T cells.

Minor points

1. On figure S1d, every labeling should be English.

Response: We feel very sorry for our mistakes and we have made correction in the revised paper.

2. The author should show FDR value for the GSEA analysis in figure 2d.

Response: Thank you and we have added FDR value for GSEA analysis in Figure 2d according to the Reviewer's comments.

3. Statistical data should be presented for the immunofluorescence assay in fig 3c.

Response: As suggested, we conducted additional immunofluorescence assay to test the co-localization of β -catenin and nuclear marker DAPI. Pearson's

correlation coefficient and Manders' overlap coefficient were used to quantify the extent of co-localization. Pearson's correlation coefficient ranges from -1.0 to 1.0, 0 indicates no significant correlation, the more the quantitative value is close to 1.0, the better the co-localization is; Manders' overlap coefficient is ranging from 0 to 1.0, which is considered to represent the true degree of colocalization⁵. As shown in Fig. 3c and Supplementary Fig. 3d, β -catenin was mainly located in the nucleus and cytoplasm in control SW480 and LoVo cells, and there had a moderate co-localization of β -catenin and nuclear DAPI in these cells. However, ectopic CD36 expression resulted in increased cytoplasmic membrane location of β -catenin and significantly weakened the nuclear co-localization between β -catenin and DAPI. c-myc was mainly located in the nucleus.

4. Input blots for SW480 is shifted in fig 4f.

Response: Sorry for our mistakes. According to the Reviewer's suggestions, we have redone the Co-IP experiments and the above-mentioned input blots for SW480 has been removed from the revised vision.

5. The authors should explain why MG132 treatment in control cells does not increase the b-catenin level in figure 5a, since b-catenin is also a protein with high turnover rate. Lysosome inhibitor should be used as a control for MG132.

Response: Thanks very much for your comments. As the data previously shown in Fig. 5a had different exposure time, so it might be hard for us to compare the relative expression of target proteins, including β -catenin. In light of the Reviewer's comments, we have exposed the gels simultaneously and compared the relative protein level of β -catenin. Firstly, as shown in Fig 4e, MG132 treatment significantly increased the protein level of β -catenin in RKO and CACO2 control cells, and only a slight increase in the cells with CD36 knockdown. Secondly, as shown in Fig 5a and Supplementary Fig 5c, MG132 treatment could upregulate the expression of β -catenin and c-myc either in the cytoplasm or in the nucleus, and the upregulations were more obvious in cells with CD36 overexpression. In addition, we further treated cells with the lysosome inhibitor 3-Methyladenine (3-MA)⁶, results revealed that 3-MA didn't affect the GPC4 protein levels, as shown in Supplementary Fig. 5b.

Special thanks to you for your good comments.

Replies to Reviewer #3 comments

Main comments:

1) In the introduction, the authors nicely claim that CD36 is essential for metastasis (or tumor progression) of glioblastoma, oral cancer, and hepatocellular cancer. However, in Pascual et al, it was also shown to be essential for metastasis in melanoma and luminal breast cancer. In addition, solid reports have shown that CD36 is essential for metastasis of serous ovarian cancer (Lengyel group), melanoma (White group at Memorial), pancreatic adenocarcinoma (a recent report showing the metastasis-promoting exosomes depend on CD36), and cervical cancer. These works should be cited and commented in the introduction and discussion.

Response: Thanks very much for your valuable comments. We have read relevant articles and revised the first paragraph of the Introduction section (Line 40-61) and the Discussion section (Line 438-458), respectively.

2) Minor comment: in Figure 2e the KD of CD36 in Caco2 cells seems to reduce proliferation rather than increase it as claimed in the text. I think this is a simple mislabelling of the figure.

Response: We are very sorry for our mislabeling mistake and thanks very much for your kindly warning. We have corrected this mistake in Fig. 2a. Sorry again for our mistake.

3) Why are Caco2 and RKO cells so apoptotic in culture? A 20% of apoptosis is an anomaly and suggests that the culture conditions are suboptimal. Is there an explanation for this?

Response: We feel very sorry for our negligence of clear indication that the apoptosis presented were under 5-fluorouracil (5-Fu) treatment. We have conducted the experiments more cautiously to compare apoptosis between cells with different CD36 status under normal and 5-Fu treated conditions, respectively. Protein expression of apoptosis markers (cleaved caspase-3, cleaved caspase-9, Bcl-XL) were also conducted with Western blot analysis. Results showed LoVo cells displayed a higher apoptosis rate after CD36 overexpression and CD36 knockdown led to a lower apoptosis rate in CACO2

cells under normal conditions, respectively. However, when treating cells with 5-Fu, overexpression of CD36 induced a significant increase of apoptosis as compared with their negative controls in both SW480 and LoVo cells, and both RKO and CACO2 cells with shCD36 showed sharply less apoptosis than their controls. Protein expression of apoptosis markers revealed consistent changes. Related pictures and statistics have been presented in Fig.2b and Supplementary Fig.2c.

4) Major point: the authors nicely show that cells that overexpress CD36 are very resistant to glucose deprivation. I think this is a very interesting result with possibly broad consequences. As primary tumors grow in size it is well known that they become hypoxic which generates areas of low glucose levels. There is substantial literature showing that hypoxic areas favor or promote metastatic spreading of tumors (whilst showing low proliferative rates of tumor cells within these areas). It should be noted that CD36+ metastatic stem cells were identified precisely as long-term quiescent cells in the primary tumor. Hence, CD36 would not favor the proliferation of primary tumor cells but would favor the metastatic spreading of cells. All of this would be consistent with the findings shown in this paper. However, the authors have not explored the effect of CD36 in CRC metastasis. I think this is quite relevant considering that a nice report by Nath and Chan has shown that high expression of CD36 correlates with metastasis in CRC patients.

I, therefore, think that the paper would significantly gain in relevance if the authors could study the possible dual role of CD36 in primary tumors versus metastasis. The results shown by the authors regarding primary tumor growth are solid. Yet I think it is necessary for the authors to study what is the role of CD36 in the metastatic potential of the CRC cell lines they are using in this study. That is, does the overexpression of KD of CD36 affect the metastatic potential of CRC cells? This

should be tested by orthotopic inoculations, intrasplenic inoculation (which should mainly result in liver metastases), and intravenous inoculation (which should primarily result in lung metastases). Providing a potential dual role of CD36 in primary tumors versus metastases would be a very interesting finding.

Response: Actually, when reading the comments provided by the Reviewer, we felt very moved and inspired, and we knew our work has been taken seriously and discussed deeply. We have read the nice report by Nath and Chan before, and it has been discussed in our revised manuscript (Discussion section, Line 484-487). In their work, the authors claimed that fatty acids uptake genes, like CD36 and CAV1, were again amplified in metastatic lesions compared to their primary tumors, which might suggest a metabolic shift to elevate the uptake of exogenous fatty acids⁷. Consistent with the analysis and suggestions by the Reviewer, in our original idea, we also thought CD36 might play a potential dual role in primary tumors versus metastasis, but in our in vitro migration and invasion assays, we found ectopic CD36 expression in SW480 significantly inhibited cell migration and invasion, while CD36 knockdown in RKO cells promoted both of them (as illustrated in following Fig.1a), as we did not do any animal experiments then, so we thought these in vitro results were not so convincing and did not show the results in our paper. As suggested, the Reviewer kindly provided three classical animal models for testing CRC metastasis, and considering the time, we chose to perform orthotopic inoculation and intrasplenic inoculation model (which mainly result in liver metastasis) to preliminary test the role of CD36 in CRC metastasis. Firstly, we performed the orthotopic inoculation model, but we failed to observe any liver metastasis in both groups (LV-RFP vs. LV-CD36) for two months, although mice with inoculation of control SW480 cells died faster than those of CD36-overexpressed group (data not shown). Then, we tried to give the mice intrasplenic inoculation to determine the extent of liver metastasis. Four weeks after injection, to our surprise, no liver metastasis was found in all the seven mice injected with CD36-overexpressed SW480 cells, while in the control group, metastatic lesions could be observed in the livers of all mice involved (Fig.1b, c). We think it's a very interesting finding and we have transfected MC38 cells with CD36-overexpressed vectors and now went on testing the role of CD36 in metastasis in C57 mice (under observation). And we think if the results were consistent with that of the nude mice, we will further explore the underlying mechanism, and if the results were opposite, it may provide us a more challenging and interesting thought about the function of CD36 in CRC metastasis.

Figure 1. a. migration (left) and invasion assays (right) were used to test the metastasis of the indicated cell lines with different CD36 expression; b. Macroscopic appearance of the livers and spleens of the mice with intrasplenic inoculation of LV-RFP (up) and LV-CD36 (down) SW480 cells; c. Representative H&E staining of the liver.

5) Quantification and statistics should be provided for the results shown in Figure

3c.

Response: As suggested, we conducted additional immunofluorescence assay to test the co-localization of β -catenin and nuclear marker DAPI under different CD36 expression. Pearson's correlation coefficient and Manders' overlap coefficient were used to quantify the co-localization. Pearson's correlation coefficient ranges from -1.0 to 1.0, 0 indicates no significant correlation and -1.0 indicates complete negative correlation, the more the quantitative value is close to 1.0, the better the co-localization is; Manders' overlap coefficient ranging from 0 to 1.0, which is considered to represent the true degree of colocalization⁵. Results showed that β -catenin was mainly located in the nucleus and cytoplasm in control SW480 and LoVo cells, and there had a moderate co-localization of β -catenin and nuclear DAPI in these cells. However, ectopic CD36 expression resulted in increased cytoplasmic membrane location of β -catenin and significantly weakened the nuclear co-localization between β -catenin and DAPI. c-myc was mainly located in the nucleus. Related pictures and statistics were presented in Fig. 3c and Supplementary Fig. 3d.

6) In line 238 it should say Figure 3f instead of Figure 5f.

Response: Sorry for our writing mistakes and it has been corrected in the revised vision.

7) The Western blots results of GPC4 in Figure 4a are rather weak. I am not doubting at all at the validity of the conclusions, but I think the authors should show clearer blots, or at least some quantification and statistics.

Response: Considering the Reviewer's suggestion, we performed additional Western blot analysis to detect GPC4 expression under different CD36 status, and revised pictures were presented in Fig. 4a.

8) The results shown in Figure 4b are likewise unclear. The reported colocalization of CD36 and GPC4 is hardly visible. Much clearer IFs should be provided and a detailed image analysis performed to demonstrate that both proteins indeed colocalize.

Response: As suggested, we increased the concentration of antibodies and conducted additional IFs to evaluate the co-localization of CD36 and GPC4 in RKO, CACO2 and SW480 cells. Results showed that CD36 signal significantly

overlapped with GPC4 on the cellular membrane and cytoplasm in all the three cells tested (mean Pearson's correlation coefficient was 0.89 in CACO2, 0.86 in RKO and 0.82 in SW480, mean Manders' overlap coefficient was 0.90 in CACO2, 0.88 in RKO and 0.86 in SW480), related the pictures and statistics were present in Fig. 4b and Supplementary Fig. 4e.

9) Results shown in Figure 6a and 6b are consistent with a role for CD36 in promoting quiescence. This comment is related to comment 4. Again, I think it is very important that the authors study the potential role of Cd36 in CRC metastasis considering the increasing literature supporting a role for this protein in metastasis. It might very well be that CD36 is also metastasis suppressive in CRC and this would indeed be an interesting finding. But whichever the results I think it is very important that the authors perform the intrasplenic, orthotopic and intravenous inoculations to test if modulating the expression of CD36 has any impact over the metastatic potential of CRC cells.

Response: Thanks for your valuable comments and I hope our response and results in comment 4) will finely answer your questions.

10) Can the authors rule out that the effect of the AAVs modulating the expression of CD36 is due to non-epithelial cells? Is the expression of CD36 affected in the colorectal stroma of mice infected with the adenovirus? I think this is a relevant issue considering the recent report in which deletion of CD36 has been shown to cause intestinal inflammation, but that the phenotype is primarily due to loss of CD36 in the endothelium rather than the epithelium (Cifarelli V et al., 2017).

Response: Thanks very much for this question. We have read the paper the Reviewer recommended and found it's an interesting finding regarding CD36 loss in the endothelial cells of small intestines⁶. In colon and rectum, early evidence showed that decreased stromal expression of CD36 was positively correlated with vascularization as a receptor of thrombospondin 1, but in our work, we found the abundant epithelial expression of CD36 on normal colonic mucosa and focused on it. In the AAV delivery mouse models, as our AAVs

have been tagged with GFP, it provided a convenient way for us to measure the transfection efficiency as well as the AAV location. By analyzing our IHC images with GFP staining (as shown below and Supplementary Fig. 7a), although some weak staining could be seen in the stroma, the strong and positive GFP-staining was mainly located in the epithelial layer of the colorectums. Therefore, we think that AAV-induced epithelial loss of CD36 might mainly account for the functional difference.

Immunohistochemistry (IHC) of GFP in the colon tissues of AOM/DSS-induced mice models (a) and ApcMin/+ mice (b). Scale bar, 100 µm (10×).

11) A recent report has shown that CD36 affects insulin signalling and glucose metabolism (Samovski D et al., Diabetes 2018). This paper should be acknowledged and discussed in the discussion section when the authors comment on the role of CD36 in glycolysis.

Response: Thanks. We have quoted relevant articles (Discussion section, Paragraph 3, Line 502-504) according to your suggestions^{8,9}.

12) The authors might also want to discuss in their paper recent papers showing that polymorphisms in CD36 affect the risk of developing colorectal cancer.

Response: Thanks. We have quoted relevant articles^{10,11} (Discussion section, Paragraph 2, Line 468-470) according to your suggestions.

Thank you so much for your valuable suggestions and warm comments.

About author:

We ranked Yi-Zhi Zhan as the third author due to his non-negligible contributions to our manuscript revision process. All the authors agree with this decision.

We tried our best to improve the manuscript and made some changes in the manuscript. These changes will not influence the content and framework of the paper, and here we did not list the changes but marked in red in our revised paper. Once again, thank you very much for your comments and suggestions.

Sincerely yours,
Yi Ding, M.D., Ph.D.

Department of Radiation Oncology, Nanfang Hospital, Southern Medical University, Guangzhou, Guangdong Province, China, 510515
Email: dingyi197980@126.com

Reference

1. Sancho, E., Batlle, E. & Clevers, H. Signaling pathways in intestinal development and cancer. *Annu. Rev. Cell Dev. Biol.* **20**, 695–723 (2004).
2. Sakane, H., Yamamoto, H., Matsumoto, S., Sato, A. & Kikuchi, A. Localization of glypican-4 in different membrane microdomains is involved in the regulation of Wnt signaling. *J. Cell Sci.* **125**, 449–460 (2012).
3. Liu, Y. *et al.* A small-molecule inhibitor of glucose transporter 1 downregulates glycolysis, induces cell-cycle arrest, and inhibits cancer cell growth in vitro and in vivo. *Mol. Cancer Ther.* **11**, 1672–1682 (2012).
4. Passmore, L. A. & Barford, D. Getting into position: the catalytic mechanisms of protein

- ubiquitylation. *Biochem. J.* **379**, 513–525 (2004).
5. Zinchuk, V. & Grossenbacher-zinchuk, O. ARTICLE IN PRESS PROGRESS IN Recent advances in quantitative colocalization analysis□: Focus on neuroscience. *Prog. Histochem. Cytochem.* **44**, 125–172 (2009).
 6. Hu, F. *et al.* Docetaxel-mediated autophagy promotes chemoresistance in castration-resistant prostate cancer cells by inhibiting STAT3. *Cancer Lett.* **416**, 24–30 (2018).
 7. Nath, A. & Chan, C. Genetic alterations in fatty acid transport and metabolism genes are associated with metastatic progression and poor prognosis of human cancers. *Nat. Publ. Gr.* 1–13 (2016). doi:10.1038/srep18669
 8. Coburn, C. T. *et al.* Defective uptake and utilization of long chain fatty acids in muscle and adipose tissues of CD36 knockout mice. *J. Biol. Chem.* **275**, 32523–32529 (2000).
 9. Fukuchi, K. *et al.* Enhanced myocardial glucose use in patients with a deficiency in long-chain fatty acid transport (CD36 deficiency). *J. Nucl. Med.* **40**, 239–243 (1999).
 10. Kuriki, K. *et al.* Increased Risk of Colorectal Cancer Due to Interactions Between Meat Consumption and the CD36 Gene A52C Polymorphism Among Japanese Increased Risk of Colorectal Cancer Due to Interactions Between Meat Consumption and the CD36 Gene A52C Polymorphism Among . 37–41 (2009).
doi:10.1207/s15327914nc5102
 11. Neumeyer, S. *et al.* Genome-wide DNA methylation differences according to oestrogen receptor beta status in colorectal cancer. *Epigenetics* **00**, 1–17 (2019).

Reviewers' Comments:

Reviewer #1:

Remarks to the Author:

The authors have thoroughly addressed all review comments with substantial new data, which significantly enhanced the manuscript. However, the applicant should include in the manuscript the metastasis data shown in the rebuttal letter.

Reviewer #2:

Remarks to the Author:

The authors have fully addressed reviewers' comments. The manuscript is recommended for publication.

Reviewer #3:

Remarks to the Author:

The authors have addressed most of my comments raised after the first submission. There was a major point though that the authors have addressed in a preliminary manner but have decided to not include in the manuscript. This related to the potential role of CD36 in CRC metastasis. Although the results they have obtained are quite solid they are opposite to what has been extensively described for CD36 in other tumor types. I agree with the authors that it is necessary to repeat these using patient-derived tumors to complement their results with established cell lines (included as a figure for the reviewers). It will be interesting to see how these results can be reconciled with the fact that the expression of CD36 is increased in CRC samples as they progress towards a more aggressive phenotype.

Of note, when the authors cite the role of CD36 in pancreatic cancer, when referring to References 15-16 my perception is that there it was shown that indeed low expression of CD36 correlated with larger tumors, but nevertheless (and quite intriguingly) this correlated with an overall poorer survival. I think this should be described better in the discussion section.

Altogether, I think the paper is now solid and I therefore recommend it for publication in Nature Communications.

Dear Editors and Reviewers :

Thank you very much for your careful work and kind consideration on publication of our paper. On behalf of my co-authors, we would like to express our great appreciation to editors and reviewers.

Thank you and best regards.

Replies to Reviewer #1 comments

Reviewer #1 (Remarks to the Author):

The authors have thoroughly addressed all review comments with substantial new data, which significantly enhanced the manuscript. However, the applicant should include in the manuscript the metastasis data shown in the rebuttal letter.

Response: Thank you very much for your kind comments and suggestions. We have added the metastasis data in our revised manuscript. Data of in vitro transwell assays and in vivo intrasplenic inoculation mice model were present in Supplementary Figure. 2e and Figure. 6c and Figure. 6d, respectively.

Replies to Reviewer #2 comments

Reviewer #2 (Remarks to the Author):

The authors have fully addressed reviewers' comments. The manuscript is recommended for publication.

Response: Thank you very much for your valuable suggestions in improving our manuscript, we sincerely appreciate your kind recommendation on publication of our paper.

Replies to Reviewer #3 comments

The authors have addressed most of my comments raised after the first submission. There was a major point though that the authors have addressed in a preliminary manner but have decided to not include in the manuscript. This related to the potential role of CD36 in CRC metastasis. Although the results they have obtained are quite solid they are opposite to what has been extensively described for CD36 in other tumor types. I agree with the authors that it is necessary to repeat these using patient-derived tumors to complement their results with established cell lines (included as a figure for the reviewers). It will be interesting to see how these results can be reconciled with the fact that the expression of CD36 is increased in CRC samples as they progress towards a more aggressive phenotype.

Of note, when the authors cite the role of CD36 in pancreatic cancer, when referring to References 15-16 my perception is that there it was shown that indeed low expression of CD36 correlated with larger tumors, but nevertheless (and quite intriguingly) this correlated with an overall poorer survival. I think this should be described better in the discussion section.

Altogether, I think the paper is now solid and I therefore recommend it for publication in Nature Communications.

Salvador Aznar Benitah

Response: Thank you Professor Salvador Aznar Benitah. Thank you very much for your valuable comments and interesting ideas in regarding our paper, as we know that the metastasis-initiating role of CD36 has been solidly reported by your team in *Nature*¹. According to your suggestions, we have discussed the role of CD36 in pancreatic cancer, and we think CD36 might also be a tumor-suppressive gene in pancreatic cancer, for its low expression is correlated with larger tumor burden and poorer overall survival and disease-free survival of patients with pancreatic cancer², but CD36 expression on immune cell promotes the engulfment and extravasation of tumor

microvesicles as well as the formation of premetastatic cell clusters to enhance liver metastasis of pancreatic cancer cells³. The difference may reflect the unique cell type-specific, context-specific and function-specific roles of CD36. In my humble opinion, I think there may exist some CD36 high expression subgroup in the whole CD36-deficiency tumors, and the tumor development and progression may result from a good cooperation of them according to different environment and functions. What's more, we found fatty acids uptake and oxidation genes, like CD36, Caveolin-1 and CPT1A, were downregulated in primary tumors but again amplified in metastatic lesions, suggesting a metabolic shift of CRC cells in metastatic sites^{4,5}. So just as you suggested, more solid data, like PDX model or immune-competent mice model, is needed to identify the exact role of CD36 in CRC metastasis, and the metabolic shift of lipid metabolism in primary and metastatic CRC also need further study to illustrate the underlying regulatory mechanisms. Once again, special and sincere thanks to you, Professor Salvador Aznar Benitah.

Reference

1. Pascual, G. *et al.* Targeting metastasis-initiating cells through the fatty acid receptor CD36. *Nature* **541**, 41–45 (2017).
2. Jia, S. *et al.* Down-expression of CD36 in pancreatic adenocarcinoma and its correlation with clinicopathological features and prognosis. *J. Cancer* **9**, 578–583 (2018).
3. Pfeiler, S. *et al.* CD36-triggered cell invasion and persistent tissue colonization by tumor microvesicles during metastasis. *FASEB J. Off. Publ. Fed. Am. Soc. Exp. Biol.* **33**, 1860–1872 (2019).
4. Wang, Y.-N. *et al.* CPT1A-mediated fatty acid oxidation promotes colorectal cancer cell metastasis by inhibiting anoikis. *Oncogene* **37**, 6025–6040 (2018).
5. Nath, A. & Chan, C. Genetic alterations in fatty acid transport and metabolism genes are associated with metastatic progression and poor prognosis of human cancers. *Sci. Rep.* **6**, 18669 (2016).